# Biomimicry in Architecture: A Review of Definitions, Case Studies, and Design Methods

**DOI:** 10.3390/biomimetics8010107

**Published:** 2023-03-07

**Authors:** Nathalie Verbrugghe, Eleonora Rubinacci, Ahmed Z. Khan

**Affiliations:** Building, Architecture and Town Planning Department (BATir), Université Libre de Bruxelles (ULB), Avenue A. Buyl 87 (CP 194/2), 1050 Brussels, Belgium

**Keywords:** biomimicry, biomimicry in architecture, BIA, sustainable design, biomimicry design approaches, biomimicry classifications, biomimicry design methods

## Abstract

Biomimicry, as a field of science, is mainly defined as a solution for design problems inspired by natural models, systems, and elements. For the built environment, using nature as a guide can enhance sustainability or even go beyond that and generate a regenerative approach. This is important in the building sector to evolve towards a sustainable and circular economy and reduce CO_2_ emissions in terms of energy-use. While several biomimicry-related keywords exist, scholars and practitioners in architecture have given varying interpretations to the term biomimicry depending on the use and goal. There has been increasing interest in biomimicry in architecture (BIA), yet the field has become more fragmented. This study aims to highlight differences and similarities through an extended literature survey and analysis that explores case studies, classification systems, and methodological frameworks related to biomimicry in architecture as a way to contribute to reduce the fragmentation in the field. To provide the necessary context and avoid confusion regarding the many concepts and terms that refer to nature-based design, biomimicry-related keywords and interpretations of the word biomimicry are first clarified. Ultimately, the discussion is an integrative effort at defining the field, and highlights the significance and impact of employing BIA in terms of sustainability and usability, as well as showcasing the opportunities for further research.

## 1. Introduction

Biomimicry is a promising emerging research field defined as a solution for design problems inspired by natural models, systems, and elements [1]. The term was coined by Janine M. Benyus in 1997 and is a junction of the Greek words ‘bios’, meaning ‘life’, and ‘mimesis’, meaning imitation [2]. When referring to design inspired from nature, terms such as biomimetics, bio-inspired, and biologically inspired are also used [3]. In 3.8 billion years, nature has created technologies equivalent or superior to those invented by humans, but with sustainable and efficient means [4]. Biomimicry as a field of science is an interdisciplinary approach and has the potential to provide sustainable solutions with the collaboration of biologists, physicists, chemists, engineers, and architects [5]. Natural systems are known for their circular resource use, intelligence, self-sustaining and energy-saving qualities [2,6]. The idea that guides biomimicry is to take inspiration from nature to help solve human and, or design problems in a more sustainable way [5]. Architecture has evolved in the last 50 years, reprioritising goals and shifting toward collaboration with other disciplines [7]. Practitioners and academics have worked to make the move from a linear economy, which depletes natural resources for the benefit of people, to a circular economy. Therefore, using nature as a guide can enhance sustainability or even generate a restorative approach [8]. Since the term was coined in 1997, biomimicry as a scientific field has emerged over the last few decades, as evidenced by the increase in publications on the topic on Scopus, illustrated in Figure 1. We surveyed this emerging interest for biomimicry in architecture, referred to as BIA in this paper, illustrated in the graph that comprises journal articles, conference proceedings, and books on the topics: ‘biomimicry’ and ‘biomimicry in architecture’. More detailed bibliometric research comprising keywords related to nature-based design is found in Section 2.2.2.

Biomimicry serves as an inspiration with a significant potential for sustainability [9], and is particularly relevant for architecture and the construction sector, which are large emitters of greenhouse gases, both during the construction and operative phases of buildings. The building sector accounts for around 40% of CO_2_ emissions related to energy use [10,11]. Using biological strategies is beneficial for various fields. To achieve sustainable and conscious architecture and minimise the overall environmental impact, buildings can be improved through the passive, efficient, and circular design principles found in nature [12]. BIA can be used to improve the external envelope by allowing more efficient ventilation, resource savings, improved thermal comfort, and the overall energy efficiency of buildings to make them all together more sustainable [10].

Architects and designers refer to nature-based design using interpretations of several bio-related keywords, such as biomimicry, biomimetics, bionics, biomorphism, ecomimicry, and organic design [13]. The field is still new and abstract, but it is already broad ranging. However, the lack of methodological clarity and a clear and consistent definition makes it challenging to find an overview and understand the true meaning and impact of employing BIA [14], and to further advance this promising field of research and practice. While there is increasing interest in biomimicry, as illustrated by our survey of articles in Figure 1, the field remains wide-ranging, but is becoming more fragmented. This fragmentation owes to the large number of terminologies found in the literature and the lack of consensus on definitions and its interpretations, practices, and approaches. Therefore, the main objective of this article is to systematically survey the terminologies and definitions, case studies, classification systems, and methodological frameworks related to BIA. The focus is to highlight their differences and similarities and point towards the need for integration as a way to contribute to reducing the fragmentation in the field. Thus, this article provides an overview of what BIA means, what the state of the art is, and how it is practiced and approached. Relevant case studies demonstrating BIA are selected and reviewed in this paper to highlight the difference between theory and reality, showcasing how actual designs align with the concept of biomimicry as found in the literature. In terms of design methods and approaches, various design frameworks for applying BIA have been developed, each aimed towards a specific topic, either within the field of biology or architecture. The terms approaches, frameworks, and methodologies are used in conjunction, all referring to a design method for incorporating BIA.

After this introduction (Section 1), and prior to the discussion and conclusion, this paper is organised into three main sections. We review biomimicry and biomimicry-related terminologies and definitions to avoid confusion or misinterpretation of nature-inspiring concepts in Section 2, which is organised into the following questions: What are the terminologies related to the field of biomimicry in terms of using nature as a concept generator, or as referred to in this article: ‘biomimicry-related keywords’?; How do they differ? and; What is biomimicry and how is it defined and interpreted among researchers and practitioners? Additionally, the biomimicry-related keywords are quantitatively surveyed through a bibliometric analysis. This expanded introductory section provides the necessary context to facilitate and bring clarity to better understand the case studies and approaches. Section 3 describes architectural case studies employing biomimicry and elaborates on the questions: What is their source of natural inspiration?; How were they designed? and; Does biomimicry always go hand in hand with sustainability? Classifications and design methods or frameworks for applying BIA are reviewed in Section 4 to answer the following questions: What are existing classification systems for BIA?; How do architects design with nature being non-experts in biology? and; What are existing design methods for integrating biological strategies into architecture? Then, Section 5 discusses the results by combining sections two, three, and four to: reflect on and highlight the differences and similarities, provide a general overview of the state of the art related to BIA, achieve a tentative consensus on the concept of BIA, and to accentuate the significance and impact in terms of sustainability and usability. In conclusion, Section 6 provides an overview of all findings and gives suggestions for future research aimed at advancing the field of biomimicry applied to architecture.

## 2. Definitions

Biomimicry, bionics, biomimetics, biomorphism, organic design, and similar terms all refer to mimicking nature somehow. However, what is the right word to use? What is the meaning of biomimicry? Where does the term come from? In order to have clarity on what biomimicry is, and particularly what it means in architecture, the related terms are discussed. Biomimicry is a term used in various scientific fields, in very different ways, but it should imply certain aspects. For BIA, there is a consensus on what already exists and what the different interpretations are among scholars. This section briefly explains the history of biomimicry and how other related terminologies are defined in the current state of the art. The first subsection focuses on the development of the word biomimicry from the 1950s. This is when the first biomimicry-related keyword appeared in an academic context. The second subsection reviews the existing terminology of both biomimicry and related keywords found in the literature, demonstrating similarities and differences with citations of various sources, an example, and an etymological analysis of each word. To quantify these aspects, a brief bibliometric study is carried out. The final subsection emphasises the significance of BIA. What does it mean and how is it used for architecture?

### 2.1. Development of the Word Biomimicry throughout History

While some academics argue that the earliest examples of biomimicry in design may be traced back to Roman Antiquity, the Mayan times, or even Greek Mythology, this article will focus on the topic’s evolution since the 1950s, when the first contemporary examples emerged. In 1957, Otto Schmitt, a bioengineer and physicist, first proposed the word biomimetics to define his new device that imitated the electrical action of a nerve [15]. Simultaneously, Jack Steele defined bionics as the science of natural systems [9]. It was reinforced in the 1960s under the heavy impact of cybernetics, a scientific field focused on regulatory systems, like the human body [5,16]. At the same time, the word ‘bionik’ was introduced, merging the German words ‘biologie’ and ‘technik’, referring to the transfer of ideas from biology to technology [17,18]. For the next 30 years, the term was decreasingly used. However, it came back strong in the late 1990s with Janine Benyus’ book *Biomimicry: Innovation Inspired by Nature* [2,18]. Benyus is also one of the co-founders of the Biomimicry Institute [19], an organisation and platform that unites the different profiles of biomimicry: designers, engineers, and biologists, to help them understand the systems and processes of nature in practice [9,20]. Janine Benyus [2] offers the most widely known definition of biomimicry in her book specifying the origin of the word, which is from the Greek words ‘bios’ for ‘life’ and ‘mimesis for ‘to mimic’ [2]. Today, according to Bajaj [18], biomimicry entails studying and modelling biological structures, functions, and ecosystems that have been modified by evolution and that are subjected to environmental testing [15,18].

### 2.2. Existing Terminology

#### 2.2.1. Keywords in Research

Biomimicry has been used by academics for approximately 70 years with the meaning we know today, under different aliases. The general definition of taking lessons from nature and using them in practical solutions to human problems is consistent throughout the literature [15]. However, every biomimicry-related keyword has a different focus. It is therefore necessary to shed light on the differences between commonly used keywords relating to biomimicry, presented in Table 1. These definitions will clarify the actual scope of and meanings within the interdisciplinary field of biomimicry. According to Verma and Punekar [21], the people involved in researching the topic have different backgrounds (biologists, architects, engineers, designers, scientists, and so on), which can explain the diverse terminologies. This diversity also results in the development of various design methods [21]. Lastly, the expression ‘nature-based’ was not included in this table because it is not specific to either architecture or technology [22]. Nature-based includes every term in Table 1, and its broad scope makes it irrelevant to this analysis. By the same token, ‘bio(-)inspiration’ and ‘bio-inspired’ are not included. To emphasise their differences and similarities, the keywords are translated in terms of how a building would mimic a flower. Lastly, each word is also separated into syllables to determine its etymology, and further accentuate the distinct meanings of each keyword.

Each term is unique, yet all fit under the umbrella of ‘nature-inspired’ design. While the author’s definitions of one term are similar, aside from some minor differences in interpretations, the keywords plainly differ. The etymological analysis and the interpretation of the flower’s translation serve to further highlight the differences. The etymology clearly exposes the distinction between the words containing ‘bio’, ‘eco’, and ‘organic’. While ‘bio’ simply refers to the description and science of life [2], ‘eco’ is derived from ‘ecology’, which is a scientific field concerned with the relationship of organisms and their environment [35]. By combining ‘bio’ and ‘eco’ with ‘morphism’ or ‘mimetic’, the focus shifts and the meaning of the word becomes clearer. ‘Morphism’ literally means imitating form, thus biomorphism imitates the form of living things. Architects are frequently inspired by nature through symbolic associations and metaphors, without taking innovation or sustainability into account [23,24]. ‘Mimicry’ or ‘mimetics’ go beyond the form and aim to represent or imitate models on a deeper level for various purposes. Biomimetics solves practical problems using an interdisciplinary approach to transform natural processes into new solutions for human systems. The sustainability impact and innovation are often also considered [14,15,18,28,40,41]. Bionics involves the word ‘nics’, derived from technique or technology, which together refer to copying or representing a function of a living creature to efficiently design technological or mechanically-driven systems [14,15,30,42]. Ecomimicry or ecomimesis imitates nature on a larger scale, specifically for the design and socio-ecological management of communities and built environments. Therefore, the local ecosystem is also assessed [18,34,36,43,44]. Lastly, ‘organic’ differs most from the others because it relates more to characterizing living things [38]. Organic design involves a threefold inspiration from nature: through symbolic associations, through the use of biological principles allowing for nature-like relationships and harmony, and through the choice of biodegradable or reusable materials [21,37].

In general, these keywords separate into three main categories, which are relevant regarding the meaning of BIA. Bionics intends to enhance mechanical systems through improved nature-inspired technologies. This is mostly found in products, but also in structural or MEP (mechanical, electrical, and plumbing) systems of buildings [42]. Biomorphism and organic design relate more to shapes and forms, and the latter bridges over to bionics as well. Ecomimicry and ecomimetics relate to the overall wellbeing of all inhabitants. Then, biomimetic(s) refers to the imitation of natural models and is commonly used as the adjective of biomimicry. Architecture that uses biomimicry as a tool is referred to as biomimetic architecture, and can be interpreted as the general term overarching all three categories depending on the end goal of the project. However, the specifics as to whether ‘biomimicry’ is the umbrella term that encompasses all these keywords, including biomimetics either as adjective or separate noun, is beyond the scope of this paper. Lastly, another keyword that is very rarely used, but is interesting for the understanding of BIA, is vernomimicry. We did not include it in Table 1 as it is not a type of biomimicry, but combines biomimicry (in the widest context of the term) and vernacular architecture. A great resemblance exists between both making this approach more accessible and understandable for architects. Hence, it is relevant to this survey. A vernomimetic approach would result in a building inspired by an aspect of a flower and vernacular knowledge solving a similar architectural challenge [45].

#### 2.2.2. Bibliometric Analysis

A bibliometric analysis was performed to provide clarity and to quantitatively portray the biomimicry-related terms. Scopus is recognised as one of the bibliographic databases with the most extensive data sources for a wide range of topics. Therefore, it was chosen to conduct a bibliometric study [46]. The state-of-the-art review was conducted in February 2023 using keywords and Boolean operators in the documents’ titles, abstracts, and author keywords, limited to publications since 1997 to 2022. All languages were selected to obtain the most representative result in terms of geographical spread. Different types of contributions were identified, such as books, conference proceedings, scientific papers, etc., using the following keywords related to BIA: “Biomimicry”, “Biomimicry AND architecture”, “Biomimetic OR Biomimetics”, “Biomimetic AND architecture”, “Biomorphism OR Biomorphic”, “Bionic OR Bionics”, “Organic AND design”, and “Ecomimicry OR Ecomimesis”. Table 2 depicts the total number of publications pertaining to the keywords and publications from 1997 to 2000, as well as the last four years, to get a sense of the augmentation.

There is a significant disparity in the number of publications using the keywords ‘biomimetic(s)’ and ‘biomimicry’. The total number of articles for ‘biomimicry’ is 2187, whereas for ‘biomimetic(s)’, the result is 75,726. This might be explained by the fact that term ‘biomimetic’ was introduced in the 1950s [15], and thus already broadly used in the academic community before Benyus coined ‘biomimicry’ in 1997. By the same token, the number of publications using the keywords ‘bionic(s)’ are also significantly higher than for ‘biomimicry’, since it was introduced in the 1960s [17,18]. Most apparent is the high number of publications for ‘organic design’, which is a term associated with the Modern Movement in architecture in the early 1900s. The words ‘organic’, ‘organic design’, and ‘organic architecture’ in the construction context were embodied by architect Frank Lloyd Wright during that time. Therefore, this keyword was already widely used in an academic context before the others [47]. In general, all have intensified in the last decades. Most publications have its origin in scientifically and technologically strong countries. Aside from some European and Asian countries, the United States and China have contributed the most concerning the keywords presented in Table 2. Moreover, concerning field distribution, most publications originate from engineering and materials science. These are related to the development of architecture, and thus have contributed to the advancement of BIA. However, biomimicry is interdisciplinary and relevant to many fields. Several articles originate from biology and medicine, making it broad, but also fragmented, prone to confusion, and ill-defined. As previously stated, the expertise of each contributor results in varied definitions and interpretations for what inspiration from nature means, specific to each field.

#### 2.2.3. Biomimicry: Common Ground and Differences of Opinion among Scholars

Biomimicry is another keyword found in the literature and differs from those presented above. Through different experiences and approaches, researchers have given distinct meanings to the word. Webster’s dictionary defines biomimicry as *“the imitation of natural biological designs or processes in engineering or invention*” [48]. ISO 18458:2015 [30], already integrates sustainability within its definition of biomimicry, defined as a *“philosophy and interdisciplinary design approaches taking nature as a model (…) to meet the challenges of sustainable development”*. A challenge for architects is the absence of a precise and shared definition and methodology among the numerous possibilities for BIA found in the literature [9]. Therefore, this section focuses on clarifying the meaning of biomimicry concerning architecture.

In several papers, biomimicry is defined as an approach to sustainable development. In the broadest sense of the concept, it emulates or takes creative inspiration from nature’s processes, impact, tactics, ideas, and systems in order to design a durable future [10,15,18,23,49]. The word ‘inspire’ refers to the designer achieving innovative design solutions [23]. It is an interdisciplinary approach, relatively new, and with a high potential for sustainable developments that involves many profiles, such as engineers, biologists, and architects [9,10,49]. Applying nature-generated design to building issues is the groundwork for environmental-friendly developments, and has already been demonstrated to have several advantages, including improving innovation, optimising resource use, and improving health [10].

However, scholars have different interpretations and ways to define biomimicry, as well as diverse arguments over what type of challenges biomimicry addresses. The overall consensus is that biological strategies are used as inspiration, but variations exist owing to its interdisciplinary nature and origin of development. The point of view and interpretation of a biologist (Benyus [2]) and of an architect (Badarnah [10]) differs. Benyus [2] defines it as an imitation of the design and processes to address human issues, whereas Badarnah [10] extends this to addressing issues through environmentally friendly developments, particular to architectural challenges. Vincent [14] diverges from this point and focusses on addressing challenges in society. Some authors further the abstract nature of biomimicry. Gruber [4] defines it as a study, not an approach, which implies a theoretical science rather than a practical solution. Vincent [14] mentions the objective of biomimicry, but not the means, whilst Zari and Hecht [50] mention both: an action, through looking at the living world to build and sustain urban environments that are robust and flexible, and an objective, to enhance the ecosystem health and regeneration. By contrast, Badarnah [10] establishes biomimicry as an approach, thereby making it more practical. Adding to that, Richter [20] and Zari [51] each present clearer definitions by explaining the source of inspiration and the way to use it. While Richter [20] states that natural principles need to be thoroughly comprehended, Zari [51] clarifies this by adding that designers can mimic an organism, its behaviour, or features of its ecosystem. The Biomimicry Institute [19] also quite concisely and clearly defines biomimicry as a practice mimicking strategies to regeneratively solve challenges. Second, the founder of that institute, Benyus [2], contradicts other sources by stating that biomimicry was new in 1997. Most other authors date the birth of biomimicry as much earlier. Chayaamor-Heil and Hannachi-Belkadi [23] present biomimicry as the newest, cutting-edge approach, and also diverge on that point by arguing that biomimicry was new in 2017. Again, taking inspiration from nature for addressing practical or aesthetical human needs has always been present and the earliest examples of modern biomimicry date back to the 1950s. Nonetheless, it was Benyus [2] who formally introduced biomimicry as a field to the academic world. Furthermore, the meaning of the word has evolved over the last years as a result of its use and applications. Third, not every researcher agrees on the type of problem that biomimicry solves. Some argue that biomimicry is more about societal and human challenges, which is only one part of the architectural design process. Gruber [4] and Badarnah [10] state that it is only used to solve architectural challenges. The Biomimicry Institute [19] mentions regeneration applicable to all scientific fields. Zari and Hecht [50] also mention regeneration and the ecosystem’s health with a focus on urban challenges, which is broader than a solely architectural question.

In general, these definitions can be explained by the scope and limitations of each paper, the expertise and sensitivity of the authors, and the overall diverse ways of interpreting the field. Nonetheless, origins differ, making it challenging to bring biologists and architects together on a common understanding of what biomimicry is and, to a greater extent, should be.

### 2.3. Biomimicry in Architectural Design

The world’s population reached 8 billion in 2022 [52]. It is widely recognised that, under the status quo, society’s hyper-consumption is not sustainable in the long term. The development of the use of regenerative resources is a way to reduce human emissions and make sure the world can sustain its population growth, particularly in the construction sector [53]. The idea behind BIA is that, since nature operates on the principles of reuse and recycling rather than producing waste energy, it is a suitable source from which to draw inspiration [18]. Additionally, biomimicry in architectural design offers a way to establish reachable growth goals that are grounded in practice. It also outlines how to do so while highlighting real-world instances that might serve as inspiration. The goal to use BIA is to make additional advancements and innovations on already used producing, generating, or capturing technologies in terms of energy and resource efficiency in general to lessen human dependence on the linear consumption of fossil fuels, which still accounts for the majority of the world’s energy use [23]. According to Bajaj [18], it has become clear over the last few years that using a nature-inspired approach to innovation and technology in architecture might mitigate some of the harmful impacts of the industrial age. Biomimicry aids in expanding sustainable thinking through principles like interconnectedness and integrating systems [18]. Due to the multidisciplinary nature of evolutionary processes, the production of design concepts includes three distinct domains: the issue, the natural world, and the solution. A design process involving biomimicry requires seeking for a biological strategy to solve a technical issue [10]. According to Badarnah [10], building envelopes that can adapt to their surroundings can increase a building’s resilience, as well as its sustainability, since they use resources more effectively and use less energy [10]. Indeed, evolution allows biological creatures to adapt by creating multifunctional and self-adaptive solutions through mutation, recombination, and selection. The outcome is a compromise that satisfies a number of needs at once [5]. As yet, BIA is effective for generating climate-responsive designs through external facades and technical systems, such as natural ventilation [12].

Thus, extracted from the analysed terminology, biomimicry is an approach that emulates natural systems to find durable solutions. In architecture, it is an interdisciplinary design method that is still quite underexplored, but can expand the designer’s realm of ideas. It is used to tackle architectural challenges to fit human requirements. It aims at increasing the built environment’s capacity for the regeneration of local ecosystems, and makes it more resilient and in some cases, incorporating evolutionary principles through adaptable designs. Nature uses energy-saving processes and closed loops with minimal waste. Biomimicry applied to architecture can satisfy numerous needs at once.

## 3. Case Studies

In the last few decades, architects have been increasingly interested in the development of architecture that is inspired by nature for ameliorating technological, aesthetic, and environmental aspects of a building. A pioneering example is the Eastgate Development Harare by Mick Pearce, constructed in 1996 [54]. We describe eleven case studies that have successfully implemented biological analogies. Built case studies of BIA are still scarce [55], even if more built and conceptual examples can be evidently found. The projects in this section are chosen for their representativeness of the use of BIA in a broad way: mimicking processes and systems or purely visual instances, using pre-existing models, sustainability aspects included or not, adopting low or high technology, enhancing structural aspects, and so on. All presented buildings are large construction projects for commercial purposes, except for one that serves as a place of worship. Table 3 provides an overview of these case studies, including the architect, location, date, biomimetic inspiration, and targeted performance. All buildings are regarded as examples of the overarching terms biomimicry and biomimetics. The last column contains the keyword(s) from Table 1 with which the architectural building can also be associated. The linked keywords were determined based on several articles, references, and the definitions of the keywords from Section 2.2.1. Additionally, vernomimicry is included in this table to accurately portray the nature of the last example.

The objective of the Eastgate Development Harare (EDH) was to be developed at a low cost, with appropriate office comfort levels, without air conditioning, and without jeopardising the aesthetics or general quality of the interior [56]. EDH consists of two office buildings linked together with a glass roof [54,57]. Each of them is topped by chimneys above internal cavities that suck out the exhaust air from the floors below. Fans take the cool air from the central atrium and inject it into the offices, in turn extracting heated air and sending it to the cavities [54]. This system is based on the termite mounds’ natural climate-controlling infrastructures [58]. It allows for night cooling, as well as thermal storage and convective air currents that regulate the temperature, thereby reducing energy costs by 65% [54,57,59]. EDH was built using clay bricks, which is the same material used by termites to build their nests [60].

The Arab World Institute (AWI) is a museum in Paris. Completed in 1987, it was envisioned as a cultural centre that celebrates the alliance between France and the Middle East. For this reason, the North façades is a mirror of the western culture, whilst the South façade celebrates the eastern culture with a modern interpretation of the ‘moucharabiehs’ [61,62,63]. High-tech mechanical diaphragms respond to light by simulating iris dilation and contraction [64]. Each module has 73 diaphragms and is programmed to perform 18 motions every day. As a result, the façade is packed with electrical components such as sensors, computers, and actuators. There are 30,000 photoelectric cells, which are light-sensitive steel diaphragms on 1600 parts that act like a camera lens, and all mechanical equipment is connected to a central computer. These are linked together in a network of metallic components that shift and rotate to create iris-like motions. Sadly, the kinetic operation of the system was stopped after only about six years due to excessive maintenance expenses, regular component replacement, and severe mechanical flaws [64,65,66].

The Eden Project (EP) is a 130,000 sqm touristic and educational attraction [67]. The main part of the park is constituted of large greenhouses, placed on different levels to keep up with the constantly shifting, former mined, landscape. It is a well-known example of BIA. The greenhouses were inspired by Buckminster Fuller’s geodesic dome, soap bubbles and honeycomb structures [68,69]. The idea of soap bubbles came from the ease with which bubbles can settle on any irregular surface with the least resistance [70]. The geodesic system is lightweight, strong, and does not require any internal support. Each hexagon of the structure is filled with several inflated layers of ETFE, a plastic polymer material lighter than glass [60]. It provides thermal insulation and furthers the structural stability of the greenhouses [57]. ETFE is however not a good acoustic insulator, tends to display condensation in-between the layers, and lets a lot of light through, hence overheating the interior of the greenhouses when exposed to high temperatures [71]. Since it is a greenhouse, the EP needs to ensure specific climatic conditions, tropical temperatures, and high humidity levels, on the interior for the fauna and flora. Each bubble is called a biome, a natural occurring flora community that occupies a major habitat [60].

The Council House 2 (CH2) houses the municipal offices of the city of Melbourne. It was the first six-star rated green building in Australia [72]. It was allegedly designed with the goal of providing a comfortable, adaptable, and stimulating working environment for its users. Additionally, the building aimed at being almost carbon-neutral and inspired a new relationship between the city and nature [73]. A study in 2013 showed that in reality the building’s energy performance is closer to four stars, scoring better than ‘average’ buildings but below its potential [74]. Nonetheless, CH2 is an example of BIA, that extends beyond sustainability, employing a regenerative effect on its surroundings [75]. Radwan and Osama [9] have found the CH2 to be very effective in its translation. The similarity to a tree filtering the air is remarkable and the building is constructed almost entirely with recycled and renewable materials [9]. Nature is employed more as a source of inspiration than as a real practical model. Indeed, the building was inspired by the notion of synergy, referring to the overlapping of systems, each of which being greater than the sum of its parts. So, for a construction, it comprises the building fabric, humans, engineering systems, energy fluxes, natural and man-made landscapes [75]. In plants, it comprises of the leaf structure, growth plane, soft body, stem, bark, dermis and epidermis, bronchi, root, and antennae, which CH2 used for inspiration [76].

The Lotus Temple (LT) is a place of worship. It is constituted of 27 free-standing petals that were constructed out of concrete and white marble coating. The LT is well known for its distinctive flowerlike shape. It was designed to replicate the beauty and symmetry of the lotus flower, which is sacred in many Eastern and Indian faiths [77]. The lotus flower is half open, symbolising the openness to all cultures, beliefs, and languages [78].

The cultural centre Esplanade Theatre (ET) is located on an area of four hectares. One of the goals of the project was to represent past and future projects, linking contemporary techniques and local traditions. The two primary buildings took inspiration from sea urchin shells and the Asian durian fruit’s spiky shell. The mimicked durian’s spikes on the roof are for aesthetical reasons, but also serve as protection from the sun composed of aluminium panels lining the sun’s path. Singapore is close to the Equator, so the sun’s path does not significantly change throughout the year. These aluminium plates are organised in a grid pattern based on curves that alter the diamond proportions. Above the grid, a second layer of aluminium panels generates the appearance of bird beaks shifting direction, flattening or rising. The panes are engineered to let in just enough light while keeping the tropical heat out, inspired by both the sea urchin shell and the durian fruit [79,80].

The One Ocean Building (OOB) displays a kinetic adaptive façade system, where lamellae can move and create patterns. This pavilion is a result of BIA inspired by the ocean [81,82]. The main entrance is surrounded by more than a hundred lamellae that mimic the opening and closing mechanism of fish grills. The lamellae are made of glass-fibre-reinforced polymers (GFRP), a material that can be moulded into a variety of dynamic designs. They are used as mobile sun-shading devices that can be programmed to adapt to changing lighting conditions, follow a predetermined dance and react to specific occurrences. Consequently, light can radiate in and out of the structure and provide views in both directions and the structure can stay closed-up. They use the Flectofin^TM^ system, a well-known biomimicry example based on the bird-of-paradise flower. This mechanical system exists thanks to the elastic properties of certain materials, and the flower itself. When adding a force, the flower is under bending and torsion at the same time. When that force is removed, it goes back to its original shape. Flectofin^TM^ has been used in a multitude of projects and is well-established. In the OOB, the asymmetrical bending is facilitated by actuators at the top and bottom [82,83].

The Gherkin Tower (GT), formally known as 30 St Mary Axe, is an example of mimicking the form of a specific organism. However, it goes beyond just literally imitating and looks at the entity of the structure. In biology, material and structure are inseparable [84]. The concept of the tower has been examined in the theory of ‘Radiolarians’, which is concerned with the way organisms build their structures in reaction to motion, either to enhance the organisms themselves or to reduce the material that hinders them [85]. The exoskeleton and round form of the Venus flower basket sponge offer it rigidity and distribute stresses from high currents. The properties of the sponge are replicated in the tower into a diagrid steel structure in the building envelope, and vectorial operations that mimic the curved shape. The tower can withstand wind forces by blowing air up through the cutaway floor openings and by making them slide around the building thanks to its cylindrical shape. This allowed for a higher construction using fewer materials [85,86].

The Sahara Forest Project (SFP) in Qatar, Tunisia, and Jordan is a large-scale project aimed at demonstrating how biomimicry can be used to address a variety of challenges with closed-loop models. This includes the passive harvest of fresh water, the transition to a solar economy, the regeneration of land, the sequestration of carbon in soils, the closing of nutrient cycles, and the employment of large groups of people. It intends to employ restorative methods to create vegetation in dry locations and reverse the trend of desertification and diminish the pressure on natural resources [1,60,87]. Since the climate does not allow for fresh water for agriculture, architects and engineers used inspiration from the fog-basking ability of the Namib Desert beetle. The Namibian Desert has about 1 cm of rainfall per year, but experiences fog events coming from the Atlantic Ocean during the mornings. This animal evolved to efficiently capture fog with its back that contains alternating hydrophobic areas, whereon droplets congregate, and hydrophilic areas, through which droplets are transported to the beetle’s mouth [42,60,88].This biological strategy translated into saltwater-cooled greenhouses, to provide suitable growing conditions and allow year-round agriculture. Fans blow desert air over seawater, allowing for evaporation, which creates humid air streams within the greenhouse and decreases the indoor temperature. The condensation of moist air using seawater-cooled pipes results in a freshwater supply for irrigation. The energy required to run the greenhouse is created in a concentrated solar power plant, where solar energy is converted into steam and utilised to power turbines [60]. At the pilot plant in Qatar, cucumbers successfully grew using half of the amount of fresh water needed under conventional conditions. SFP complies with many of the criteria of an ecosystem: nutrient-rich agriculture, no lasting pollutants, diversity, energy gains through solar power, and regenerative as an entity [1].

The Homeostatic Façade (HS) comprises a self-regulating façade system that automatically responds to changing external conditions such as daylight and temperature fluctuations. HS uses the homeostatic principle of muscles, which allows organisms to control internal parameters such as temperature [11,89]. The system is made up of an engineered strip that is placed inside the cavity of a double-skin glass façade. The ribbon is formed of dielectric rubber compounds, which are polymeric materials that can be polarised by passing an electrical current through them, causing them to stretch. These materials are flexible and use minimal energy. The silver layers/electrodes conduct electricity through the material and can reflect light. The charge in the silver layer creates motion using a sensitive actuator when the external circumstances change. The dielectric electroactive polymer is wrapped around a flexible polymer core to form an artificial muscle. Increased charges force the elastomer to expand, causing the core to bend and the elastomer to pull to one side. As a result, the facade closes [57]. Ultimately, the interior conditions are controlled by balancing heat losses and gains, hence also conserving energy [11,89].

Cairo’s Gate Residence (CGR) uses principles from vernacular architecture, bioclimatic design, and biomimicry. The project combines traditional wind catcher techniques of Egypt, Iran, and the Gulf area, as well as an extensive use of renewable energy systems to save 50% of energy demand, uses the solar cycle, prevailing wind directions, endemic plan species, geothermal energy, and more to make the design as passive as possible. It aims at transforming the city into an ecosystem, and the district into a forest [90]. ‘Malqaf’ are traditional wind catchers found in Egyptian constructions from 1300 BC. It is a shaft rising high above the building with a unidirectional opening facing the prevailing wind. It catches the wind flowing over the building, and internal air ducts push it down into the spaces of the house. In the larger buildings, like mosques, the ‘maqlaf’ and a higher tower on the other side of the building for hot air escape work in conjunction, just like termite mounds and the EDH. ‘Modern’ buildings from the 1960s to today ignore the vernacular ‘maqlaf’, demanding large cooling and energy-intensive systems [91]. The CGR adopts the ‘malqaf’ in the format of nine large chimneys, called ‘megatrees’, that guide the airflow in three directions: downward using direct wind entry, upward using a wind-assisted temperature gradient, and upward using a solar-assisted temperature gradient [90,92].

These case studies were chosen for their representativeness and online availability of information. Small-scale projects for domestic purposes integrating biological strategies are less common as biomimicry is still an emerging tool and often implies an initial higher cost, ergo, less cost-effective at smaller scales. Cruz and Hubert [57] provide additional large construction projects, but also small pavilions employing biomimicry. Nevertheless, the case studies display the difference between theory and reality and showcase how actual designs align with the concept of biomimicry as found in the literature. In theory biomimicry is presented as an effective approach for sustainable or even regenerative solutions, as highlighted in the definitions of Badarnah [10], the Biomimicry Institute [19], and Zari and Hecht [50], but this is not always included in practice. Adopting BIA can be driven for enhancing sustainability or solely for innovation [93], and if the end goal of the designer is to create an efficient, durable structure, simply mimicking nature is not enough. Henceforth, aside from natural principles, the need for being conscious of every choice is crucial in terms of, for example, materials and their provenance, structural efficiency, and bioclimatic design.

## 4. Classifications and Design Methods for Biomimicry

Relevant classifications and methodological frameworks for architectural purposes are reviewed in this section to unfold their similarities and differences. The distinction between classifications and design methods can be confusing because several examples are applicable to both. While they can be used to classify buildings employing design principles of biomimicry, they also serve as design frameworks for translating biological features into technological implementations. Therefore, when referring to a ‘design method’ for incorporating biomimicry, this includes approaches, frameworks, and methodologies.

### 4.1. Classification Systems for Biomimicry in Architecture

The classification systems for bio-inspired architecture are mostly categorised by the mimicked biological features. Benyus [2] distinguishes three levels of biomimicry: organism, behaviour, and ecosystem. These are used and are applicable to all fields. A design can mimic some parts of an organism, the response of an organism in its context, or a function of an ecosystem [2]. Zari [94] added an additional dimension to biomimicry for researching biological analogies tailed towards architectural applications, which are: form, material, construction, process, and function. The same author depicts the differences between each level and aspect with the example of a building mimicking termites [94]. Table 4 provides an overview and explanation of the different levels a building can mimic a natural feature, including a tentative classification of the case studies discussed in Section 3. Buildings are inherently complex, and some can have more than one strategy that uses biomimicry, not all on the same level. For example, CGR functions on the level of an ecosystem (it stores water and uses solar power) and also on the behaviour level (careful orientation, form, and use of natural ventilation) [60]. Overall, the organism level is primarily inspiring for a building’s form, shape, or structure, whereas the behaviour is for studying the interaction of the building with its surroundings. Inspiration for ameliorating the urban metabolism and how local organisms and the built environment interact on a larger scale is usually found at the ecosystem level [95].

The table assembles the crossroad between the design aspects and natural levels of biomimicry. The organism-level case studies that are designed based on the form of that organism are the Esplanade Theatre and the Lotus Temple. The ET, the Gherkin Tower, and the Homeostatic façade are also constructed in the same manner as their respective mimicked organism. Indeed, the ET has spike-like elements that protect the inside from the sun, the GT has a round shape and a lattice-like exoskeleton, mimicking the shape of a Venus flower basket [60], and the HT presents a ribbon that inflates and deflates, mimicking muscles. The Arab World Institute works as an organism through the iris’ dilatation for light control, and the GT as well through the dispersion of stresses in different directions thanks to its shape [60]. Because of its cylindrical shape, the GT also functions as the flower in a broader context, in the sense that it disperses stresses and reduces wind (or current) forces [60]. Then, the behaviour-level case studies mimic organisms with its context. The Eastgate Development Harare uses the same material as termite mounds, which is clay, mimicking the organism (material). The One Ocean Building is constructed in the same manner as the bird-of-paradise flower, with specific hinges, and mimics its opening and closing process. The EDH and the Cairo Gate Residence also imitate the natural ventilation process of termite mounds [60]. Lastly, the ecosystem-level case studies resemble the ecosystem, and the same three buildings also function like termites, or like the flower by controlling indoor parameters. The Eden Project and the Sahara Forest Project are constructed as an ecosystem: mimicking the shape and structural efficiency of the bubbles for the EP and copying how the Namibian beetles collect water for the SFP. The EP, SFP, CGR, and Council House 2 have similar processes to their inspiration. The EP works like an individual bubble (collecting large amounts of sunlight), the SFP uses both salt water and solar power, and CGR and CH2 extensively use renewables and closed-loop resource systems. The SFP also functions like an ecosystem thanks to seawater loops. This classification system is convenient for efficiently discovering eligible and functional biological analogies. However, it lacks multifunctionality and clarity, since it can create confusion because there is no exclusive answer. Biomimicry-inspired buildings can be distributed over several aspects, especially when multiple biological strategies are used. Furthermore, as a designer, it is difficult to correctly classify projects following that system with a limited knowledge of biology. An expansion of the database and the creation of a universal classification system through collaboration with biologists and architects is highly recommended.

A different approach for categorising BIA is by using a catalogue in the format of a checklist comprised of identifying biological strategies, the link to biology, the design process utilised, the outcome, and the level of adaptability. As for the previous classification, a collaboration with a biologist is beneficial for identifying the biological principles. The list, however, is mainly developed for architects through the use of understandable terminology [28,57]. Presenting this as a method for classification, the link to biology and targeted performance, for example, can be combined, as depicted in Table 5. Aside from ‘animalia’, ‘plantae’, and ‘other’, the original checklist integrates ‘fungi’, ‘bacteria’, and ‘protozoa’ as biological strategies [57]. Examples can be found in [96], but these are not integrated into the figure, since no described case studies found its inspiration in these areas. Furthermore, these are primarily used for enhancing materials’ performance, which is not within the scope of this paper.

These are levels of biomimicry based on resource regulations, which are the most important for climate-responsive designs. The Lotus Temple can thus not be integrated into this classification since the mimicry level is merely for aesthetic purposes. Again, further development of case studies showcasing BIA could enlarge the classification system.

Lastly, Ilieva et al. [97] developed the Biomimicry for Sustainability framework, which assesses the amount of impact of BIA. Case studies employing biomimicry are categorised along two dimensions on the *x*- and *y*-axis, respectively. The first dimension (*x*-axis) depicts the scope for mimicking nature in relation to sustainability. Four categories are identified being biomimicry for: ‘innovation’, ‘net-zero optimisation’, ‘societal transformation’, and ‘biosynergy’. The sustainability impact of the latter is assumed to be highest, whereas that of innovation is nearly zero. The *y*-axis represents whether it is a fixed or flexible mimesis, which does not contribute to the sustainability impact. When nature plays an active role in the design process, literally translating and integrating nature, it is referred to as fixed, whereas flexible entails a more passive interpretation [97]. In this regard, the case studies of Section 3 are tentatively depicted in Table 6, employing this framework.

The case studies classified as biomimicry for innovation barely contribute to sustainability, but focus more on novelty and economical objectives [97]. Even though AWI has a high potential for being sustainable through its climate-adaptive façade, it falls into that first category because of the overuse of mechanically driven components. Net-zero optimisation is concerned with, first, the efficiency in terms of materials (the GT qualifies because of its nature-inspired structure reducing the use of materials), and second, the performance in terms of energy use (EDH qualifies due to its integration of a passive ventilation system). Then, the SFP aims to encourage the transition from a linear to a circular economy, thus using biomimicry for societal transformation, and uses many passive strategies. Therefore, the SFP is also an example of biosynergy displaying a regenerative impact on nature and is concerned with all living organisms. The CGR is the only project where nature plays an active role in the impact of sustainability through the extensive use of green areas, thus qualifying as a flexible mimesis. Note that the type of mimesis has no influence on sustainability impact, but is merely a new way of categorisation [97]. The framework also holds the design process into account. However, due to the lack of indirect information from the designers of the case studies, differences in opinion for categorisation are inevitable.

### 4.2. Design Methods for Biomimicry

A design process, either in engineering or architecture, usually begins with the question of what the design should be. The evolution and adaptation of nature to fit into its environment have proven to be a viable source of inspiration for innovative applications [98]. The study of biomimicry is not simply copying external characteristics of organisms. The design approach involves an in-depth investigation for mimicking biological mechanisms on a physiological, morphological, and behaviour level [12]. According to Benyus [2], nature has developed nine design principles, referred to as ‘Life’s Principles’: the use of sunlight; only use required energy; form to function; recycle all resources; reward cooperation; focus on diversity; demand local expertise; curb excesses from within; and tap the power of limits [2]. These can be summarised into two prevailing ideas as a base for BIA. First, every organism has the desire to reproduce and maintain itself, called ‘autopoiesis’ or ‘conativity’. Second is the path of least resistance in order to achieve conative goals [99]. These nature-based principles serve as an efficient measuring tool. As a designer, shifting the question to what the design needs to do and seeking biological analogies for guidance has the potential to result in genuinely sustainable or even regenerative solutions [98]. Various design approaches to integrate and translate natural principles for solving or meeting human challenges have been developed over the years, which are elaborated on in this section and illustrated in Figure 2.

#### 4.2.1. General Design Methods for Biomimicry

The basis of any biomimetic research is the examination of a specific biological phenomenon. In general, two biomimetic ‘Research by Design’ approaches as a design field can be distinguished according to their process sequences, each comprising six steps [3,12]. The difference between the approaches is the starting point for development. When the research starts with a promising biological discovery with a possible technical application, it is called a bottom-up approach (or ‘solution-based approach’ or ‘biology to design’). When a designer starts with a specific technical question where the solution is looked for in biology, a top-down approach is employed (or ‘problem-based approach’ or ‘design to biology’) [9,100]. Additionally, a third approach is defined as the extended top-down and explores various biological analogies for a clear-cut technical problem through an iterative process. This extended top-down approach is driven by the returning question of: ‘What if nature has an even better solution?’, requiring more time and research, but with the potential for creative and out-of-the-box outcomes [3]. Designers mostly employ the top-down approach [60]. In both main approaches, the biological feature is abstracted from the organism and translated into a technical implementation. A designer’s traditional design approach differs from a biomimicry-based one in its concept generation. Biomimicry is interdisciplinary, whereas designers usually look for solutions within their field of expertise. Most designers are not educated in biology, therefore all design methods benefit from a collaboration with biologists and existing databases during the discovery phase for creative and sustainable findings [99]. Mostly, collaborations aid in the development of literally or metaphorically translating biological strategies, depending on the mimicking level. Moreover, the abstraction phase is always the most challenging one for non-experts in biology [3,12]. What to search for?; Where to find it? and; How to identify an interesting biological strategy? are common obstacles that designers face [101].

**Figure 2 biomimetics-08-00107-f002:**
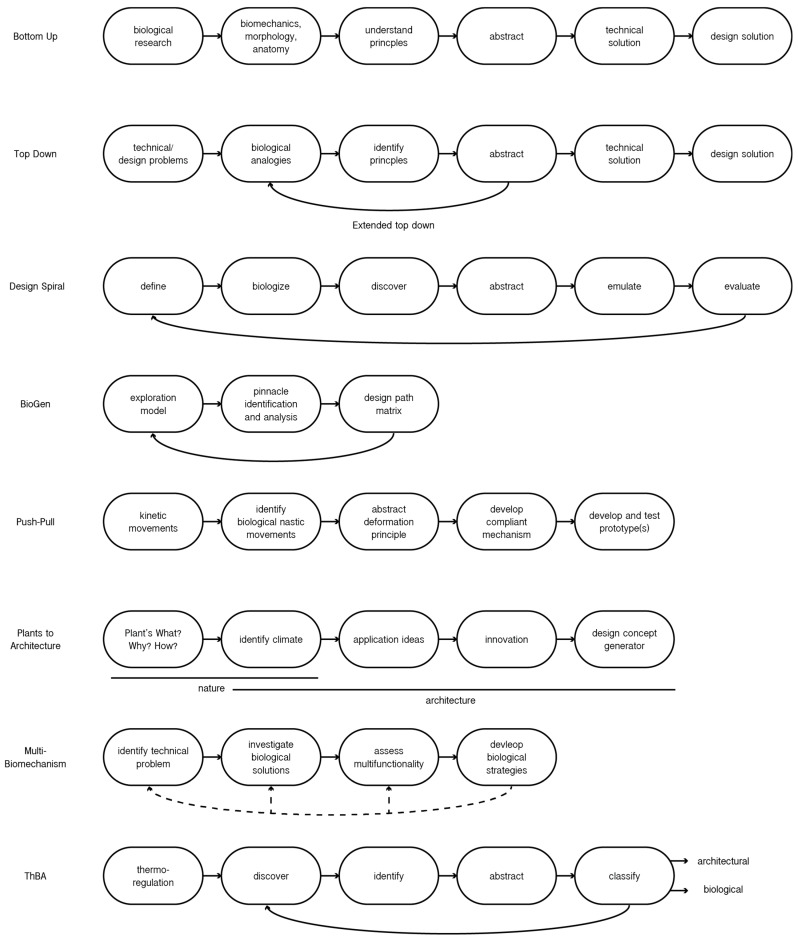
Simplified stages of biomimetic approaches described in Section 4.2. From top towards bottom [adapted from]: Bottom Up [3], Top Down [3], Design Spiral [19], BioGen [102], Push-Pull [103], Plant to Architecture [104], Multi-Biomechanism [105], and ThBA [106].

Biomimicry 3.8 is an online platform accessible to everyone that provides education, inspiration, methodologies, and other biomimicry-related information. The Challenge to Biology Design Spiral is an approach developed by the Biomimicry Institute for designers from all scientific fields. Nature employs a reiterative design process, commonly known as evolution, towards the most efficient mechanisms adapted to its environment and the path of least resistance. The Design Spiral guides designers through this process, employing the following steps: define, biologise, discover, abstract, emulate, and evaluate [19]. In order to facilitate the discovery and abstraction phases, Biomimicry 3.8 offers an online database, called AskNature, which functions as a field guide to the natural world. The database contains over 1600 biological strategies classified through a framework, the ‘Biomimicry Taxonomy’ that links the biological adaptations in relation to functional challenges [107]. The platform provides a significant aid during the initial phase of the design process. Nonetheless, only a limited number of biological strategies are provided and collaboration with a biology expert is still recommended to broaden the range of possibilities. Kuru et al. [105] also argue that the limitation of the platform, and respectively, the Design Spiral, is the lack of assigning multifunctional properties of organisms to multifunctional challenges [105].


#### 4.2.2. Design Methods for Biomimicry in Architecture

Methodological approaches are crucial for architects to effectively integrate inspiration from nature at an early stage, preferably before or during the concept generation. Design methods for BIA are reviewed in this section and are chosen, as they are commonly used in architecture and literature, and their differences cover many design concepts. The BioGen (Biomimetic Design Concept Generation) and the ThBA (Thermo-Bio-Architectural Framework) are approaches that employ an iterative top-down methodology. The Push-Pull method is linear and uses both the top-down and bottom-up approaches. Plants to Architecture uses Push-Pull to create kinetic structures solely inspired by plant movements. Multi-Biomechanism can be both linear or iterative, and uses both main approaches. This section concludes with briefly elaborating on the BioTRIZ methodology and on regenerative design approaches. Undoubtedly, more frameworks exist, but they extend beyond the scope of this research.

The Biomimetic Design Concept Generation or BioGen is a framework developed for standardising dominant biological strategies for related functional challenges. Prior to applying the methodology, a specific technical problem is usually defined. Again, the question of what the design needs to do is raised. The concept generation is achieved by employing tools during the preliminary design stage. First, the exploration model identifies viable biological strategies parallel to the imposed technological challenge. Subsequently, the best-performing natural ecosystems and organisms are isolated, which are referred to as pinnacles. Second is the pinnacle analysis, where an in-depth investigation uncovers the biological function to mimic according to its morphology, physiology, and behaviour. Lastly, the design path matrix is the abstraction phase. A preliminary design is conceived by merging all strategies. If the outcome is not as desired or further improvement is needed, the process is reinitiated [102]. The methodology has been effectively employed for the creation of a conceptual shading device for envelopes inspired by plant movements tracking sun radiation [108]. As applicable to the previous approach, the methodology lacks a well-defined process for abstracting the organisms, especially for non-experts [109]. Furthermore, when using various pinnacles, it is crucial to maintain a clear overview, and avoid conflicts between different strategies.

In a more recent publication, Badarnah [10] expanded this to facilitate the abstraction phase and address multifunctionality through the principle that ‘form follows environment’. Organisms tend to solve various issues through one strategy, whereas humans mostly focus on solving one single issue with one strategy. The theoretical methodology proposes a framework to enhance the multi-regulation aspects of four environmental factors (heat, light, water, and air) for biomimicry-based, adaptive facades. By linking the environmental factors to specific pinnacles, possibilities for multi-regulation are assessed. A biological strategy, for example, that addresses multifunctionality in the natural world is wrinkles on the skin. The irregular surface of the skin: (1) provides a large surface area to hold moisture; (2) allows evaporation; and (3) creates shade. This pinnacle has an impact and the potential to regulate all four environmental factors. An architectural application for facades is cooling through external cladding [10]. The methodology seeks to promote the use of environmental factors during the initial stage of the design process, but again, the proposed framework could benefit from a focused database of biological strategies and practical applications [105].

The Thermo-Bio-Architectural Framework or ThBA was developed by Imani and Vale [106], and provides a framework for architects that seek a solution for nature-based thermoregulation techniques. ThBA begins with a pre-defined technical issue, being thermal performance. It is developed for bridging the gap between architecture and nature by providing an immediate relevant biological example to mimic the problems raised in terms of building energy use. The framework employs a cross-disciplinary approach linking architectural and natural features. For instance, thermal principles in architecture are made understandable for biologists, and vice versa. The biological analogies are linked through a design-by-analogy design process in a systematic manner. The framework identifies, classifies, and categorises biological strategies. Thermoregulation in buildings can be passive or active, and researchers have discovered a parallel connection with the natural concepts. Parameters for energy-efficient buildings were matched to these strategies, serving as the architectural side of the framework [12,106].

Schleicher et al. [103] developed the Push-Pull methodological framework, which employs a combination of the bottom-up and top-down approaches for the development of kinetic structures inspired by plant movements. The name is derived from ‘pulling’ a biological aspect for ‘pushing’ technological development. The approach is most suitable for external shading device systems to protect the interior from the sun through spatial adjustments. The first design phase is preferably conducted by biologists to identify biological strategies that respond to external stimuli (or nastic movements) through movement. The exploration and simplification of the mechanism and rearrangement of the motion components can contribute to a large design freedom, instead of directly mimicking the organism on a morphological level. Then, the deformation principle is translated into a bio-inspired compliant mechanism, usually through computational aid. Important to convey is that plant movements are mostly triggered by elastic deformation and flexible members, whereas conventional mechanisms use technical hinges in combination with stiff parts. This should be regarded and mimicked as much as possible to obtain a functional biological product. The last phase is to test and validate developed prototypes [103]. The proof of concept has been validated for the development of Flectofin^TM^ [110].

While the frameworks ThBA and Push-Pull focus on a specific functional architectural aspect that needs to be solved by looking at the natural world, the focus can also be on a specific group of organisms. Lopez et al. [104] developed a design approach for the development of adaptive architectural envelopes by studying plants. A strong analogy exists because both, plants and conventional buildings, lack the possibility for movement to conquer environmental factors. The concept generator collects data and maps plant strategies to guide the transfer from biology to architecture. First the plant’s adaptation is analysed according to the three questions: What?; Why?; and How? These findings are then assessed for technical implementation according to three concepts: application ideas and adaptability, possible innovation with challenges and benefits, and design concept generation with technical implementation and features [104]. The methodology offers a clear framework and aids in the translation. However, a hurdle is finding viable organisms, so developing a database of plant adaptation strategies would be beneficial for the use of this framework. Furthermore, it seems to lack insights in analysing biological features. Therefore, a combination with the Multi-Biomechanism approach could be helpful, which focuses more on developing a wide spectrum of possibilities for a biological strategy and is described next [105].

Kuru et al. [105] developed the Multi-Biomechanism approach, which aims to achieve multifunctionality in adaptive building skins. The framework has four stages. Every approach starts with either identifying the biological principles or the technical issue. In this case, the step is subdivided into identifying the location’s climate and the building performance analysis together with identifying functional requirements. Second is the selection of appropriate biological solutions, which are systematically classified. The third step is crucial for achieving multifunctionality. Through the principle of hierarchy, the organism is studied on all levels, beginning with its ecosystem, and ending with its atom. The organism’s morphological, physiological, or behavioural adaptations in a heterogeneous structure are also identified and examined. The biological analogy is developed by translating the working mechanisms to functioning and applicable materials, geometries, and configurations for the façade system together with actuation methods for climate-adaptability [105]. For the successful use of this approach, designers require an extended understanding of biological strategies.

BioTRIZ is a well-known problem-solving method used to create a bridge between biology and engineering. The goal is to turn biological vocabulary into a technological solution, its main mechanism is based on revealing conflicting requirements between biology and engineering, and creating a win-win resolution [105,111,112,113,114]. TRIZ is more focused on simple and direct systems. However, buildings are complex, and the tool might not be able to take the multiple interactions among building components into consideration. For that reason, it might not be appropriate for architectural design [105,115].

Biomimicry can serve as a tool for developing regenerative designs. The basis for regeneration is to understand the location and create for the wellbeing of all present life [116]. Over the last few years, regeneration has gained attention to rethink the green building sector [117]. By understanding the principles of the local ecosystem, a building could contribute and function within the system toward a neutral environmental outcome instead of diminishing the ecosystem’s health. Zari [118] established six ‘ecosystem services’, which serve as a framework with key parameters for designing in a built environment with a regenerative approach. When seen as a system, the built environment may offer a habitat for all organisms, contribute to soil fertility, purify environmental resources, regulate the climate, produce renewable energy, and locally collect water [118]. This is rather linked to ecomimicry. Thus, the combination of an approach integrating BIA and an ecomimetic approach, could genuinely increase regeneration.

The frameworks described give an overview of how biomimicry can be used in practice during the design phase. However, as also concluded from the case studies, this is not sufficient to guarantee a building’s sustainability, and the latter must be designed with other bioclimatic strategies in mind [94]. For instance, the design goal could be to have a symbolic association with a particular element of nature, like in the Lotus Temple. Goals can be aesthetic, symbolic, technical, structural, based on the energy performance of the building, and so on. The presented design methods are similar to Benyus’ design spiral, which is a general approach and can be used in various fields. The difference among the methods resides in their scope and focus. For example, ThBA aims to tackle a specific problem, the thermal performance and thermo-regulation of buildings in particular. For advancing the field of BIA, a combination of carefully selected frameworks, depending on the end goal, together with a unified classification system and a general expansion of the database of biological strategies, is the most important step. Furthermore, a design method for transdisciplinary research enhancing communication, and ultimately, collaboration is lacking in all approaches [119].

## 5. Discussion

The meaning and impact of biomimicry in architecture (BIA) has shown different facets throughout this paper. The analysis of the nature-inspired terminologies that relate to BIA of Section 2 provided an overview and helped clarify the discipline, but also revealed its fragmentation. In terms of how natural imitation is as an approach, each keyword has a different focus. While some focus on symbolic associations, others are more focussed on innovation and others include sustainability as the main focus. However, nature applied to architecture can sometimes have a utopian tendency, describing it as a solution that solves societal and architectural problems, and leads humans to a sustainable future in the construction sector. Is sustainability the main goal of biomimicry and other related terminologies? What is the driving force of designers instigating the development of such sustainable or regenerative projects? Experts have argued that rather than intrinsically being driven for the wellbeing of all creatures, designers are mostly driven by environmental policies, rating systems, and benchmarks [120]. Cruz et. al. [40] argued that biomimetics and biomimicry are distinct, in that the latter approach is especially focused on producing sustainable solutions, whilst the former does not need to meet that condition. Biomimetics is defined as a creative method based on the observation of biological processes. It is not required to accomplish long-term goals [40]. However, biomimicry and biomimetic are interchangeably used in many papers for semantic reasons. In the current language, or in research in general, biomimetic is often used as the adjective of the word biomimicry, and not the word biomimetics. This makes it difficult to distinguish between both terms, even if their meaning can be distinctively interpreted. After the examination of biomimicry in practice in Section 3, it can be argued that the sustainability-aspect is indeed not always addressed. For instance, Council House 2 used recycled timber for its outer facade and focuses on the all-around sustainability of every aspect of the buildings [9]. This is something missing in the other case studies. By contrast, the Arab World Institute showcases a clear biological analogy, but also an excessive amount of technology usage which encountered mechanical issues [66]. This is not only the case for the use of technology. The Lotus Temple, for instance, allegedly used natural forms in its design, but also used an excessive amount of concrete [77]. Rather than having investigated nature for a more resource-efficient structural system or materials, the designers used nature for its shapes and symbolic associations without contributing to sustainability prospects.

All case studies from Section 3 are associated with the terms biomimicry and biomimetics, as they mimic natural phenomena from either plants or animals (except for the Eden Project, which relates most to the flexibility and strength of soap bubbles). Concerning the keywords identifying the case studies, every example can be associated with a more precise keyword referring to technology, shapes, wellbeing, or the vernacular. The Lotus Temple, which uses the lotus flower as a symbol of openness, and to a lesser extent, the Esplanade Theatre, which echoes the local durian fruit by using spikes as heat regulators, purely imitate the form on an organism level. This is biomorphism, as the building looks like an organism. In this paper it is assumed that biomorphism fits under the more generic term of biomimicry. This is supported by Zari’s classification system [94], and by Sommese et al. [11], who classified the Lotus Temple as an example of BIA. However, according to the Biomimicry Institute [25] and Chayaamor-Heil and Hannachi-Belkadi [23], biomorphism is very different from biomimicry, since it does not specifically seek to solve problems, but solely imitates nature for aesthetic or symbolic purposes, and biomorphic architecture should thus not be classified as biomimicry [23,25].

Many described case studies employed bionics for technological advancements. This is the case for the Arab World Institute, which looked at the iris of the eye and how it expands or contracts in response to visual stimuli; for the One Ocean Building, which incorporates the bending mechanism of the bird-of-paradise flower; for the Sahara Forest Project, which mimics the Namib Desert beetle’s fog-basking ability; for the Homeostatic Façade, which uses muscle-like processes to control visual and thermal parameters; and lastly, also for the Eden Project and the Gherkin Tower, mimicking an organism with an adapted technology based on specific shapes for structural purposes. The Eastgate Development Harare and the Cairo Gate Residence are linked to organic design, mimicking a certain process, as well as being physically alike. Both buildings imitate the process of the natural ventilation system seen in termite mounds. EDH looks physically similar because of the shared use of clay, whereas, for the CGR, it is because of the resembling interior shapes. Ecomimicry and ecomimesis can be slightly distinguished according to its etymology showcased by the case studies. The Sahara Forest Project and the Eden Project relate to ecomimicry, working like and integrating into the local ecosystem. The Council House 2 is an example of ecomimesis and aims to imitate the local ecosystem through a multitude of biological strategies. The harvesting of fog contributes to the overall green regeneration of the area, and the EP’s interior works like an ecosystem, while the particular shape of soap bubbles helped the building’s adaptation to the uneven ground in situ. Finally, the Cairo Gate Residence uses vernacular inspiration (traditional wind catchers) to cool down the interior, which is based on termite mounds, and thus relates to vernomimicry.

After the overview of the different classification systems and design methods in Section 4, including a step-by-step exploration to increase accessibility and their ease of use, it becomes clear that the abstraction and translation of each method are difficult to exercise without the collaboration of diverse scientific profiles. It was also showcased that not all frameworks are aimed at meeting human requirements. However, potentially, a combination of various frameworks and methodologies enhances the sustainability aspects. This requires more preliminary work of studying suitable frameworks before starting on the design of a product or project. Section 4 began by presenting three classification systems. The first is based on Benyus’ definition of levels or scales in nature: ecosystem, behaviour, and organism, which is subdivided into design elements by Zari as: form, material, construction, process, or function. Despite the system lacking multifunctionality and refinement, it is especially good at rapidly classifying and providing an overview of projects or elements regarding biological strategies. The second uses checklists, asking where the inspiration was taken from in nature and what the targeted performance in architecture is. It was created for architects without much knowledge in biology. Third, the approach Biomimicry for Sustainability assesses the impact of a biomimetic product or building. There are two dimensions: the scope of mimicking nature in relation to sustainability, and whether it is a fixed (literal) or a flexible (abstract) mimesis. A combination of the three would be the most effective for evaluating a design. This would provide an inclusive overview of the links between natural strategies and the built project, a precise evaluation of the methods used, and their results, and the actual impact on sustainability. All projects could thus be classified based on more than one dimension and the link between these dimensions could be highlighted, for instance, through the influence of using a level of Zari in relation to the assessed sustainability impact. However, its full development does not fit within the scope of this review.

Three general and five design methods integrating nature into architectural design were reviewed in the second part of Section 4. The top-down approach is linear and starts with a design problem before turning to biology for answers. The bottom-up approach is linear and starts with a biological process before looking at what human problem it could solve. The design spiral is iterative, based on the evolution process of nature. The first reviewed method specific to architecture was the BioGen approach, which is an iterative top-down methodology that standardises biological strategies for technical problems. Then, the Push-Pull method is linear and uses both the top-down and bottom-up approaches by pushing a biological aspect. The Plants to Architecture framework uses Push-Pull to create kinetic structures inspired by plant movements. The Multi-Biomechanism approach can be linear or iterative, and uses both main approaches for multifunctional problems for developing nature-inspired adaptive skins. At last, ThBA is an iterative top-down method for solutions regarding thermoregulation in architecture. Even if all these approaches and classification systems exist, it is difficult to find one that is integrated, shared, and generally agreed upon. Regarding the case studies and the design methods, it is worth noting that nature entails a multidisciplinary approach, as it addresses several issues with one single element. This aspect is only included in one case study, CH2. By contrast, the Multi-Biomechanism and the extended BioGen approaches aim to tackle several architectural issues using a single biological strategy, whereas other frameworks do not mimic this natural characteristic. In order to design multidisciplinary aspects within a project, one must employ the adequate framework, as they do not all have the same output.

Common to all design approaches, translating natural features to functional applications comes with various obstacles. Khoja and Waheeb [45] proposed an approach combining nature as a generator for design and vernacular architecture to bridge the gap between architecture and biology. In this study, a vernacular house in Cairo (Egypt) is compared to the natural ventilation strategy of a termite mound, showing great similarity and success regarding indoor thermal comfort. This example could have been integrated in the development of the Eastgate Development Harare because, in practice, EDH operates differently from the actual termite mound. While a ‘vernomimetic’ approach still entails challenges, vernacular architecture provides examples that are easier to understand by architects [45]. The Cairo Gate Residence uses these principles by employing regional knowledge via the ‘maqtab’ and a biomimicry-based approach for meeting the 21st Century’s demands in terms of the environmentally friendly use of energy [90,91].

By taking all the presented aspects of BIA into account, it can be concluded that biomimicry, as an approach, emulates natural systems to find potential durable solutions, depending on the methodology and ultimate design goal. Biomimicry is an interdisciplinary process demanding collaboration with biologists. This aspect is also included in Benyus’ original definition [2]. In theory, it is straightforward. In practice, or in methodological terms, not so much. Moreover, further applied research, together with the development of a comprehensive and unified approach for architects, could increase the built environment’s capacity, increase regeneration, and make it more resource efficient, resilient, and adaptive. A design approach integrating biological strategies is not sufficient to guarantee a building’s sustainability, and the latter must be designed with other bioclimatic strategies in mind. Nature works based on energy-saving processes and closed loops with minimal waste, and by using multifunctional frameworks and studying biological analogies on several scales. Biomimicry applied to architecture can satisfy numerous needs at once.

## 6. Conclusions

Although there is a growing interest in biomimicry in architecture or BIA, the field is becoming increasingly fragmented. Therefore, this article reviewed and summarised topics related to BIA through an extended literature survey and analysis. This article, with the examination of nature-inspired terms and their interpretations, offered the fundamental context for this survey and avoided misinterpretation. Related to that, practical applications in the form of case studies, classification systems, and methodological frameworks were described. Thereby, the differences and similarities between topics related to BIA were highlighted to emphasise the necessity for unification as a strategy to eliminate dispersion in the field, which is the main goal of this paper. While many articles have focused on either the history, bibliometric research, case studies, or specifically developed methodological frameworks (for example, for building skins or particular technologies), no article collectively reviewed all related aspects to unravel and understand the true meaning of employing biomimicry for enhancing architecture. The essence of what biomimicry is applied to architecture is evolved by confronting the more practical aspects of biomimicry with the theories behind it. First, a brief history of biomimicry and how associated terminologies came to be as we know them today was provided. Many keywords exist in relation to the general term biomimicry using nature for different purposes. All the biomimicry-related terms relate to nature-inspired design, and there is a certain consensus on what biomimicry is, but they differ in their approach to biological inspiration and the challenges they aim to solve. While sustainability is insisted upon in several biomimicry-related terminologies and within the definitions of biomimicry among scholars, and the case studies highlighted the potential of that said sustainability, some cases focused more on addressing other aspects. Moreover, several keywords dominate the present-day literature, some pointing toward symbolic associations in terms of shapes; others to processes and systems, some are specific to technological use, or in conjunction with vernacular architecture. Biomimicry differs from most words by being an all-round approach, instead of a single process. Most scholars agree on two common elements: taking lessons from nature and using them in practical solutions to human problems. However, interpretations of BIA change depending on the use and the researcher. Biomimicry remains quite an abstract field that requires further research in general. The case studies illustrate the gap between theory and practice. Specifically, it highlighted the difference between using biomimicry for sustainability or in spite of it. All are based on either plants or animals, except for the Eden Project, which relates most to soap bubbles. Council House 2 comes closest to being sustainable, even regenerative, favourably contributing to its environment, and does not solely mimic one process but a multitude of them. The Arab World Institute and the Lotus Temple, on the other hand, demonstrated that a biomimetic design approach does not always lead to a sustainable design. Indeed, the designer must be conscious of their choices throughout the entirety of the project, and, if the goal of a construction is to be sustainable, use biomimicry to that end. The described frameworks showcased technicalities, making clear that the field is still new and can create confusion. Nonetheless, the classifications and methodological frameworks for BIA helped give a rounder meaning to biomimicry, focussing on specific methods, and on how to translate nature into practice. The classification system of Benyus’ ecosystem, behaviour, and organism, which are subdivided into form, material, construction, process, and function by Zari, are also used in the frameworks for classifying biological strategies. Generally, all frameworks entail: identifying biomimetic aspects, translating the principle, and assessing the level of adaptability through prototyping. A last step that should be included is assessing the impact on sustainability, whether it is for products, façade structures, or an entire construction project. All frameworks implied that a lack of knowledge in biology decreases the impact of biomimicry in architecture in terms of creativity, innovation, and sustainability. This study is limited to reviewing a subset of existing definitions, case studies, and frameworks that are relevant to BIA and does not invent an inclusive design method or framework. Further research could include the development of a new classification system that integrates the strengths of those already present in the literature and reviewed here, as well as new methodologies for BIA, tailored to a specific architectural challenge. The development of a unified database of biological analogies and new classification systems and design methods could also help make the field more accessible to designers and researchers. Further studies could also determine the impact of collaboration between biologists, architects, and engineers. From the analysis conducted in this article, biomimicry appears to be the encompassing umbrella term for architecture. However, the field is still rather new and remains quite abstract, requiring more research in general to uncover the specifics, true meaning, and potential of BIA. Therefore, promoting awareness, training programs, education, and further collaboration among scientists can stimulate the use of this practice for sustainable development. In conclusion, the building sector needs to urgently shift to conscious architecture and reduce its contribution to global warming. Nature is a viable source of inspiration for novel applications in architecture, and certainly for creating climate-adaptive and resource-efficient technologies. In practice, however, biomimicry is not widespread and lacks a generalised and shared methodology. Furthermore, biomimicry does not always imply a high level of sustainability, whereas it has the potential to go much further and generate regenerative solutions.

## Figures and Tables

**Figure 1 biomimetics-08-00107-f001:**
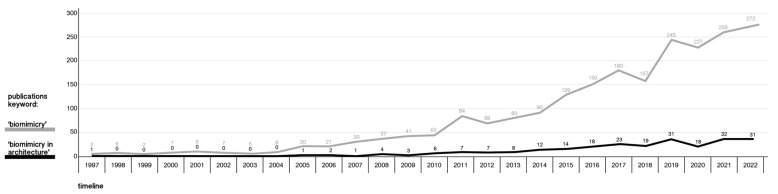
Publications (Scopus) on the topic of biomimicry using the keywords ‘biomimicry in architecture’ (black) and ‘biomimicry’ (grey), from 1997 to 2022.

**Table 1 biomimetics-08-00107-t001:** Definitions of biomimicry-related keywords in the format of quotes of several authors, all employing nature as a generator for design in the broadest sense of this concept.

Keyword	Definition(s)
Biomorphism /Biomorphic	Chayaamor-Heil and Hannachi-Belkadi [23]: *“(…) architects frequently use nature as a source for unconventional forms and for symbolic association. Nature is the ultimate in performance-orientated design so it is no wonder that attention should finally be paid to its processes. Rather than just symbolic or form, biomimetic architecture should be concerned more on aspects of how we process our design and what if our design could be a positive impact to the environment as a whole.”* Cazzaro [24]: *“Form is the result of forces located inside the matter, as it also happens in the generation of a living organism. For this reason the discourse on the creation and graphic representation of these objects can be approached from the point of view of biomorphism, a feature that can be found in similar artefacts on several levels: from the figurativity of the glass animals to the almost abstract shape obtained from the self-organising matter through its intrinsic forces.”* Biomimicry Institute [25]: *“(…) designs that visually resemble elements from life (they “look like” nature)”* Bernett [26]: *“Biomorphism mimics natural forms and patterns. It is commonly critiqued for its lack of adherence to biological principles, resulting in designs that do not necessarily perform better or that are sustainable. However, the psychological aesthetic impacts of natural forms should not be overlooked.”*
	The building looks like a flower. bio, from Greek ‘bios’ life [2] + morphism, *“from Greek ‘mophos’ having the form of and ‘ismos’ state or condition”* [27].
Biomimetic(s)	Sharma and Sakar [15]: *“Interdisciplinary cooperation of biology and technology or other fields of innovation with the goal of solving practival problems through the function analysis of biological systems, their abstraction into models and the transfer into and application of these models to the solution.”* Vincent et al. [14]: *“Biomimetics operates across the border between living and non-living systems. And (…) the reason for looking to nature for solutions is to enhance technical functions.”* Cruz et al. [28]: *“The main significant difference between ‘biomimetics’ and ‘biomimicry’ is that the approach referring to the latter tends to be specifically focused on developing sustainable solutions; the former does not have to fit that requirement. Like ‘biomimetics,’ ‘bioinspiration,’ defined as a ‘creative approach based on the observation of biological systems,’ (…) does not have to meet sustainable goals.”* DeLuca [29]: *“(…) refers to the technical translation and realization of functional strategies used by biological organisms or systems in Nature. The goal for biomimetics is to create incredibly novel radical technologies that outperform or even displace existing technologies and, in doing so, result in financial reward.”* ISO 18458:2015 [30]: *“interdisciplinary cooperation of biology and technology or other fields of innovation with the goal of solving practical problems through the function analysis of biological systems (…), their abstraction (…) into models (…), and the transfer into and application of these models to the solution”*
	The building imitates a process of a flower. bio, from Greek ‘bios’ life [2] + mimetics, *“representing or imitating something”* [31].
Bionic(s)	Vincent et al. [14]: *“(…) coined by Jack Steele (…). He defined it as the science of systems which have some function copied from nature, or which represent characteristics of natural systems or their analogues.”* ISO 18458:2015 [30]: *“technical discipline that seeks to replicate, increase, or replace biological functions by their electronic and/or mechanical equivalents”* Marshall [32]: *“Bionics is a term invented (…) to describe the prospective field involving copying imitating and learning from Nature. Since then the term in English has become focused upon mimicking human tissues and organs for biomedical purposes (thus it might be thought contiguous with biomechanical engineering).”*
	The building imitates a flower with an adapted (mechanical) technology. bio, from Greek ‘bios’ life [2] + nics, from technics [17,18].
Ecomimicry/Ecomimesis	Bajaj [18]: *“Emulation of ecosystems in design”* Marshall [33]: *“Ecomimicry is an innovation in innovation. It’s an as yet experimental technique to bring a Green quality to the process of innovation–so that novel technologies (and novel practices) can emerge in an ecofriendly manner with non-expert input. The broad idea is that the natural world may serve as inspiration for innovative ideas.”* Marshall [32]: *“Ecomimicry is the practice of designing socially responsive and environmental responsible technologies for a particular locale based upon the characteristics of animals, plants, and ecosystems of that locale.”* Winter et al. [34]: *“(…) we define ecomimicry as a strategy for developing and managing cultural landscapes, built upon a deep understanding of the structure and function of ecosystems, that harnesses ecosystem processes for the purpose of balancing and sustaining key ecosystem services, rather than maximizing one service (e.g., food production) to the detriment of others. Ecomimicry arises through novel, place-based innovations or is adopted from elsewhere and adapted to local conditions.”*
	The building works like a local flower and integrates into the local ecosystem. eco, from Greek ‘ecos’ environment and man’s relation to it [35] + mimicry, *“mimicry involves the deceptive imitation of social and political models in order to reach a certain aim”* [36]. In other words, pretending to imitate models to reach a goal. + mimesis, *“mimesis captures endeavours to imitate well-established models of social and political organisation”* [36]. In other words, attempting at rightly imitating models. It is a way of thinking.
Organic Design	Verma and Punekar [21]: *“Organic designs exhibit a very close resemblance with nature especially in terms of form and structure. Form giving for organic design is a special class of design problem that involves the use of inspiration and analogies from nature for creative problem-solving.”* Champion [37]: *““Organic architecture” is often taken to mean buildings constructed by reusable or biodegradable materials. Sometimes it is an accolade (or insult) for forms inspired by nature. A third way is to propose that architecture can be designed to coax engagement from its inhabitants, allowing them to appreciate the thought, design and care which created it.”*
	The building looks like a flower and imitates a certain process. organic, *“from Greek ‘organikos’ of or pertaining to an organ, serving as instruments or engines”* [38] and design, from Latin ‘designare’ to make, shape [39].

**Table 2 biomimetics-08-00107-t002:** Publications from 1997 to 2022, 1997 to 2000, and 2019 to 2022 from the keywords: “Biomimicry”, “Biomimicry AND architecture”, “Biomimetic OR Biomimetics”, “Biomimetic architecture”, “Biomorphism OR Biomorphic”, “Bionic OR Bionics”, “Organic AND design”, and “Ecomimicry OR Ecomimesis” on Scopus.

Keywords	Number of Publications		
(TITLE-ABS-KEY)	Total (1997–2022)	1997–2000	2019–2022
Biomimicry	2187	17	1011
Biomimicry architecture	248	1	119
Biomimetic/Biomimetics	75,726	1008	27,319
Biomimetic architecture	2823	32	1070
Biomorphism/Biomorphic	1044	21	233
Bionic/Bionics	14,328	138	6116
Organic design	106,107	3194	40,351
Ecomimicry/Ecomimesis	14	0	6

**Table 3 biomimetics-08-00107-t003:** Overview of case studies described in this section depicting the Name, Abbreviation, Reference, Architect, Location, Construction Date, Biological Analogy, Targeted Performance, and Keywords of the: Eastgate Development Harare (EDH), Arab World Institute (AWI), Eden Project (EP), Council House 2 (CH2), Lotus Temple (LT), Esplanade Theatre (ET), One Ocean Building (OOB), Gherkin Tower (GT), Sahara Forest Project (SFP), Homeostatic façade (HF), and Cairo Gate Residence (CGR). The biological inspiration of a project can be from Animalia, Plantae, or Other, depicted in the table with ^1^, ^2^ and ^3^, respectively.

Name, Abbreviation, Reference	Architect	Location	Date	Biological Analogy	Targeted Performance	Keyword
				Animalia ^1^, Plantae ^2^, Other ^3^		Biomimicry, Biomimetics
Eastgate Development Harare, EDH, [54,56,57,58,59,60]	Mick Pearce	Harare, Zimbabwe	1996	Termite mounds ^1^	Thermal comfort, Air quality	Organic design
Arab World Institute, AWI, [61,62,63,64,65,66]	Jean Nouvel	Paris, France	1987	Iris of the eye ^1^	Thermal and visual comfort	Bionics
Eden Project, EP, [57,60,67,68,69,70,71]	Grimshaw Architects	Cornwall, UK	2001	Soap formation ^3^	Mechanical resistance, thermal comfort, water regulation	Bionics; Ecomimicry/Ecomimesis
Council House 2, CH2, [9,72,73,74,75,76]	Mick Pearce, Rob Adams	Melbourne, Australia	2006	Synergy of a plant/tree ^2^	Thermal and visual comfort, Air quality	Ecomimicry/Ecomimesis
Lotus Temple, LT, [77,78]	Fariburz Sahba	New Delhi, India	1986	Lotus flower ^2^	Symbolic association	Biomorphism/Biomorphic
Esplanade Theatre, ET, [79,80]	Michael Wilford, DP Architects, James Stirling	Singapore, Singapore	2002	Sea urchin shells ^2^ Durian fruit ^2^	Thermal and visual comfort	Biomorphism/Biomorphic
One Ocean Building, OOB, [81,82,83]	soma	Yeosu, South Korea	2012	Birds-of-paradise-flower ^2^ (Flectofin^TM^)	Thermal and visual comfort	Bionics
Gherkin Tower, GT, [84,85,86]	Foster + Partners	London, UK	2003	Venus flower basket sponge ^2^	Mechanical resistance	Bionics
Sahara Forest Project, SFH, [1,42,60,87,88]	Max Fordham CE, Exploration Architecture	Sahara Desert, Qatar, Tunisia, Jordan	2017	Namibian Desert Beetle ^1^	Water regulation	Ecomimicry/Ecomimesis; Bionics
Homeostatic Façade, HF, [11,57,89]	Martina Decker, Peter Yeadon	New York City, USA	2012	Muscles ^1^	Thermal and visual comfort	Bionics
Cairo Gate Residence, CGR, [90,91,92]	Vincent Callebaut, Injaz Development, K+A Design	Cairo, Egypt	2019	Termite mounds ^1^	Thermal comfort	Vernomimicry; Organic design

**Table 4 biomimetics-08-00107-t004:** Case studies from Section 3 classified according to the levels of biomimicry. (Left: Natural levels of biomimicry in terms of ecosystem, behaviour, or organism; top: Design aspects mimicking form, material, construction, process, or function. List of acronyms: EDH (Eastgate Development Harare), AWI (Arab World Institute), EP (Eden Project), CH2 (Council House 2), LT (Lotus Temple), ET (Esplanade Theatre), OOB (One Ocean Building), GT (Gherkin Tower), SFP (Sahara Forest Project), HF (Homeostatic façade), and CGR (Cairo Gate Residence). (Adapted from [60,94]).

		Design Aspects of Biomimicry				
		Form	Material	Construction	Process	Function
**Natural** **levels of** **biomimicry**	**ecosystem**	Resembles an ecosystem EP	Used the same material as in an ecosystem	Assembled similarly as an ecosystem, growing complexity over time EP; SFP	Works similarly as an ecosystem EP; CH2; SFP; CGR	Functions similarly as an ecosystem SFP
	**behaviour**	Looks like it is made by an organism	Made from the same material an organism uses EDH	Assembled in the same way an organism would OOB	Works in the same way as the home of an organism EDH; OOB; CGR	Functions in the same way as if an organism would have built it EDH; OOB; CGR
	**organism**	Looks like an organism ET; LT	Made from the same material as an organism	Made in the same way as an organism ET; GT; HF	Works in the same way as an organism AWI; GT	Functions in the same way as an organism GT

**Table 5 biomimetics-08-00107-t005:** Case studies from Section 3 classified according to the checklist of biomimicry for architects. (Left: Inspiration from animalia, plantae, or other; top: Targeted architectural performance in terms of thermal comfort, visual comfort, acoustic comfort, air quality, mechanical resistance, or water regulation). List of acronyms: EDH (Eastgate Development Harare), AWI (Arab World Institute), EP (Eden Project), CH2 (Council House 2), LT (Lotus Temple), ET (Esplanade Theatre), OOB (One Ocean Building), GT (Gherkin Tower), SFP (Sahara Forest Project), HF (Homeostatic façade), and CGR (Cairo Gate Residence). (Adapted from [57]).

		Targeted Performance: Architecture					
		Thermal Comfort	Visual Comfort	Acoustic Comfort	Air Quality	Mechanical Resistance	Water Regulation
**Inspiration from nature: biology**	**animalia**	EDH; AWI; HF; CGR	AWI; HF		EDH		SFP
	**plantae**	ET; CH2	ET; OOB; CH2		CH2	GT	
	**other**	EP				EP	EP

**Table 6 biomimetics-08-00107-t006:** Case studies from Section 3 classified according to the ‘Biomimicry for Sustainability’ framework. (Left: flexible or fixed mimesis; top: Biomimicry for innovation, net-zero optimization, societal transformation, or synergy). List of acronyms: EDH (Eastgate Development Harare), AWI (Arab World Institute), EP (Eden Project), CH2 (Council House 2), ET (Esplanade Theatre), OOB (One Ocean Building), GT (Gherkin Tower), SFP (Sahara Forest Project), HF (Homeostatic façade), and CGR (Cairo Gate Residence). (Adapted from [97]).

		Biomimicry for			
		Innovation	Net-Zero Optimisation	Societal Transformation	Biosynergy
**Mimesis**	**flexible**				CGR
	**fixed**	AWI; LT; HF	OOB; GT; EDH; ET; EP	SFP	CH2; SFP

## Data Availability

Data available within the article.

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
