# Peer review of "Biomimicry in Architecture: A Review of Definitions, Case Studies, and Design Methods"

_biomimetics, 2023, doi:10.3390/biomimetics8010107_

Round 1
Reviewer 1 Report
There are substantial flaws to this paper.
1. I`m sorry, the outline of the paper did not convince me that the definition of term "biomimicry" should be changed and that proposed changes have not been part of initial definition. There are good leads in this paper, indeed - there is confusing and misleading use of terms regarding this topic, but I`m afraid the proposed definition will not make it clearer. With the good start and valuable effort presented partially in the paper - I would recommend resubmitting the article with redefined research question. Maybe it could be the attempt to classify wider spectrum of examples under the different terms used and highlight more thoroughly the differences and similarities between the terms based on examples.
2. Nevertheless the list of used references is appropriate, there are not enough examples to base the analysis in neither of sections. The weakest in my opinion was the analysis of definitions (2.2.2.). Are these really alternative definitions or paraphrased text to avoid plagiarism that were analysed in this chapter? Elaboration of figure 1 should be based on much more deeper literature review in the scope of defined goals.
3. Inconsistency with the mentioning of biomimetics. In the 49th line authors say that biomimetics is not included in the scope of the study, but it looks further in the text that authors use biomimetics as a synonym to biomimicry.
Please find my comments in the added pdf file. I hope that my comments will help you to restructure your paper, add the depth to all sections (more sources) and I strongly suggest renaming the paper and redefine the research question. The paper touches important topic in the biomimicry field.
Author Response
Please also see the uploaded pdf file.
Response to Reviewer 1 Comments
I`m sorry, the outline of the paper did not convince me that the definition of term "biomimicry" should be changed and that proposed changes have not been part of initial definition. There are good leads in this paper, indeed - there is confusing and misleading use of terms regarding this topic, but I`m afraid the proposed definition will not make it clearer. With the good start and valuable effort presented partially in the paper - I would recommend resubmitting the article with redefined research question. Maybe it could be the attempt to classify wider spectrum of examples under the different terms used and highlight more thoroughly the differences and similarities between the terms based on examples.
We agree and thank the reviewer for the suggestion for the redefined research question. The main objective of this paper has been amended to reviewing differences and similarities in order to highlight this as a way to contribute to reduce the fragmentation in the field. Therefore, the abstract (lines 15-19) and the introduction (lines 73-100) have been changed accordingly. In addition, the discussion and the conclusion have been reworked.
Nevertheless the list of used references is appropriate, there are not enough examples to base the analysis in neither of sections. The weakest in my opinion was the analysis of definitions (2.2.2.). Are these really alternative definitions or paraphrased text to avoid plagiarism that were analysed in this chapter?
We agree with the reviewer that section 2.2.2. (now 2.2.3) can be expanded to insist on their differences. Therefore, we have provided an example of how a building would mimic a flower according to the interpretation of biomimicry of the different scholars in Table 5. Additionally, this has also been done for Table 1 in section 2.2.1., where the example is in accordance with the biomimicry-related keywords. We believe that by providing an easy-to-understand example, the differences and similarities are clearer. We would also like to emphasise that the text in Table 5 are indeed different interpretations of several scholars and that these are paraphrased and referenced.
Elaboration of figure 1 should be based on much more deeper literature review in the scope of defined goals.
We agree that Figure 1 could be expanded through a deeper literature review. It was initially intended for a brief demonstration of the rise of publications since Janine M. Benyus coined the term. As per your recommendation, google scholar is indeed not appropriate, and thus we decided to use Scopus for the bibliometric analysis. Additionally, we expanded the bibliometric analysis and added a new part to Section 2 (2.2.2) to further this bibliographic study through a bibliometric analysis of the terminology presented in Table 1. Section 2.2.2. Bibliometric analysis now presents three tables: Table 2. Number of publications; Table 3. Publications by geographical area; and Table 4. Publications by subject area. We believe that this additional analysis enhanced the quality of the literature review and provides an additional dimension for highlighting the differences among the keywords.
We would like to add that the Venn diagram (Figure 2) has been reworked together with an expansion of the paragraph to explain this better (lines 160-178).
Inconsistency with the mentioning of biomimetics. In the 49th line authors say that biomimetics is not included in the scope of the study, but it looks further in the text that authors use biomimetics as a synonym to biomimicry.
We apologise for our lack of consistency. We do use ‘biomimetics’ in Section 2 to highlight the difference between biomimicry-related terms, as well as in the discussion section. Therefore, ‘biomimetic’ has been amended to ‘biomimicry’ where needed throughout the entire paper.
Please find my comments in the added pdf file. I hope that my comments will help you to restructure your paper, add the depth to all sections (more sources) and I strongly suggest renaming the paper and redefine the research question. The paper touches important topic in the biomimicry field.
We would like to thank the reviewer for sparing the time to write so many useful comments and providing an additional pdf. We would like to add that we have provided three more case studies. We hope that by amending the premise of the publication, by reworking sections 2 to 5 and the conclusion according to the revised research question, it is now ready for publication.

Reviewer 2 Report
Introduction
There is no link between the research objectives and the problem that has been highlighted in the introduction. Also, it is difficult to discern the scope of the research. What types of buildings are the focus of the study? The current version of the manuscript needs to be improved in terms of problem existence, its importance, identify objectives to achieve as per the problem statement, and outline the steps that will be conducted to attain the objectives.
Literature Review
The literature review section seems like a summary of everything remotely connected to the topic and end abruptly. Also, the sections need to be linked with the research objectives that the authors have identified in the introduction. Please focus the review on showing why the works so far could be more extensive and how the current research might prove better. In other words, the main arguments are missing or limited at best. As a result of not conducting a critical analysis of previous works, it is difficult to discern the research gap the manuscript aims to achieve.
Conclusion
The conclusion highlights a different research problem than the one mentioned in the introduction. Please be consistent with the aim or goal of the manuscript. The authors need to summarize the whole paper in a few non-repeated sentences from each section: the problem's existence and problem importance, lack of previous contributions to justify the current paper, what the manuscript has done to bridge the gap, how it was done, and what the findings are, and especially what the findings mean. The contribution of the findings to the body of knowledge and practice is missing in the current version of the manuscript. There is no mention of the limitations of the study. Finally, there is no direction for future research (what should be explored next), and why is that the best course of action?
Author Response
Please also see the added pdf file.
Response to Reviewer 2 Comments
Introduction. There is no link between the research objectives and the problem that has been highlighted in the introduction. Also, it is difficult to discern the scope of the research. What types of buildings are the focus of the study? The current version of the manuscript needs to be improved in terms of problem existence, its importance, identify objectives to achieve as per the problem statement, and outline the steps that will be conducted to attain the objectives.
As reviewer 1 pointed out as well, we agree and have amended the research question to reviewing differences and similarities to highlight this as a way to contribute to reduce the fragmentation in the field. Therefore, the abstract, the introduction, the discussion, and the conclusion have been reworked accordingly to answer the new main goal.
As for the case studies, we apologise for our lack of clarity. As we have explained in the first paragraph of Section 3, we have chosen these buildings for their representativeness of the use of biomimicry in architecture in a broad way. As per the recommendation of reviewer 1, we have provided three additional examples. Moreover, we have expanded the first paragraph of Section 3 to better justify our choice of selected case studies (lines 324-335).
Literature Review. The literature review section seems like a summary of everything remotely connected to the topic and end abruptly. Also, the sections need to be linked with the research objectives that the authors have identified in the introduction. Please focus the review on showing why the works so far could be more extensive and how the current research might prove better. In other words, the main arguments are missing or limited at best. As a result of not conducting a critical analysis of previous works, it is difficult to discern the research gap the manuscript aims to achieve.
As reviewer 1 also highlighted and as partially mentioned in the previous answer, we agree and have therefore changed the premise of the paper. We believe that by reframing the main objective, the literature review is now in accordance with what the manuscript aims to achieve. Nevertheless, all sections have been expanded to better highlight the differences and similarities.
Conclusion. The conclusion highlights a different research problem than the one mentioned in the introduction. Please be consistent with the aim or goal of the manuscript. The authors need to summarize the whole paper in a few non-repeated sentences from each section: the problem's existence and problem importance, lack of previous contributions to justify the current paper, what the manuscript has done to bridge the gap, how it was done, and what the findings are, and especially what the findings mean. The contribution of the findings to the body of knowledge and practice is missing in the current version of the manuscript. There is no mention of the limitations of the study. Finally, there is no direction for future research (what should be explored next), and why is that the best course of action?
We thank the reviewer for pointing this out and for giving insightful comments. The conclusion has been amended accordingly. While the discussion now entails the findings of the article, addressing the differences and similarities and combining all sections, the conclusion follows the structure as suggested, including an expansion with directions for further research.
We would like to thank the referee for the time spend reviewing our paper. We hope that the changes made meet your expectations.

Round 2
Reviewer 1 Report
Dear authors, thank You for the effort
I see how You have changed the scope, and added information, but paper still lacks scientific novelty and added value is not very high. I still find that this paper could be improved so that it is a paper worth reading.
There are lately published review papers on biomimicry in architecture:
A critical review of biomimetic building envelopes: towards a bio-adaptive model from nature to architecture
Design processes and multi-regulation of biomimetic building skins: A comparative analysis
Every next article has to bring something new to the field.
In this paper it is not quite clear, what is the "take home" message. That there are wording-based similarities and differences between what different authors write referring to boibased designs? Are these substantial differencies / similarities? How could it be measured / quantified?
The weakest part of this paper is the 2.2. subchapter. Analysis of definitions in Table 1 (what I still believe to be paraphrased versions of the same text) looks like authors interpretations, it lucks indicators, criterions to ground this interpretetion. The amount of literature sources is not enough to highlight the differences (if any) between the terms. Does all the papers in bibliographic analysis comply with the identified definitions (autors leads us to believe that there are substantial differences) or among those again is a spectrum of interpretations and at the and it all overlaps? Definition of biotechnology referes to just one source for the biggest group in bibliometris analysis. And that only one is a bachelor's thesis. There is no depth to this table.
Fig.2. lucks consistency as well. Presented scheme is not supported by thorough literature analysis. I said it already in previous version - from this scheme one could conclude that all the examples of biomorphism is biomimicry but it is not - by far. Most biomorphism examples are not biomimicry. The same goes for other terms in this scheme - there is no added value to this scheme - just compilation of "related" words. Even if it refers to another literature source, it is misleading. I would avoid this scheme until authors can add significant depth to it. Biomimicry institue has made it much more clear Venn diagram, maybe you could elaborate based on that. https://biomimicry.org/what-is-biomimicry/.
I value the effort, however bibliometric analysis is performed very poorly.
1. Proper syntax of the search including "and", "or", "not" (or other) and sensitivity to the word endings has to be prepared and described.
2. Biomimetic and Biomimetics - why is this worth looking separately to these two words? Most probably all the "biomimetic" papers are included in "Biomimetics" search
3. Biomorphism search has only 22 results because the search is not accurate - it does not include varieties in the word endings - biomorphic, for instance
4. Organic design 100k results- how those are relevant to the topic?
5. Ecomimicry gives <10 results and is excluded from bibliometric analysis, but is included in the fig 2
6. table 4 highlights the lack of deep analysis - it just shows that there is a lot of mentioning of some key - words across different fields. Because the search is not tuned, it is too wide
Table 5 - the same as for table 1 - it seems like interpretations not substantially different definitions.
In section 3 table 6 - keywords are again authors' interpretations because there are other literature sources where other keywords are used for the same example.
If authors want to keep analysis of definitions it HAS to be grounded in some kind of quantifiable parameters or otherwise prove that there is substantial difference.
Section 4 is better
Good luck!
Author Response
Please also see the added pdf file.
Response to Reviewer 1 Comments
Dear authors, thank You for the effort
I see how You have changed the scope, and added information, but paper still lacks scientific novelty and added value is not very high. I still find that this paper could be improved so that it is a paper worth reading.
There are lately published review papers on biomimicry in architecture:
A critical review of biomimetic building envelopes: towards a bio-adaptive model from nature to architecture
Design processes and multi-regulation of biomimetic building skins: A comparative analysis
Every next article has to bring something new to the field.
In this paper it is not quite clear, what is the "take home" message. That there are wording-based similarities and differences between what different authors write referring to boibased designs? Are these substantial differencies / similarities? How could it be measured / quantified?
We thank the reviewer for providing these interesting articles, which are integrated into the research ([11] and [41]). The focus of this article is on quantitative research, pointing out the differences and similarities through the various interpretations and definitions, case studies, and design methodologies. As for the terminologies (Section 2), we believe that further quantification of the words is not needed for the scope of this paper, which is to briefly introduce the nature-based terminologies in order to avoid misinterpretation in sections 3 and 4. We would like to remind the reviewer that the objective of this paper is to have a better understanding of what biomimicry in architecture (BIA) means within the landscape of all the different terms and interpretations, and their applications and approaches. In order to facilitate and empower the reader for sections 3 and 4, we truly believe that Section 2 contributes to this. Nonetheless, we would like to thank the reviewer for pushing us to enhance the quality of this paper and reflect on the choices made. We have amended various parts of Section 2, according to the objective, to genuinely contribute to highlighting the differences in a qualitative, and to a lesser extent a quantitative manner, explained in the answers below.
The weakest part of this paper is the 2.2. subchapter. Analysis of definitions in Table 1 (what I still believe to be paraphrased versions of the same text) looks like authors interpretations, it lucks indicators, criterions to ground this interpretetion. The amount of literature sources is not enough to highlight the differences (if any) between the terms. Does all the papers in bibliographic analysis comply with the identified definitions (autors leads us to believe that there are substantial differences) or among those again is a spectrum of interpretations and at the and it all overlaps? Definition of biotechnology referes to just one source for the biggest group in bibliometris analysis. And that only one is a bachelor's thesis. There is no depth to this table.
We thank the reviewer for these insightful comments. We have amended Section 2, with the focus to provide the necessary context and facilitate the reader for the next sections. The changes made regarding Table 1 and its discussion are explained in the format of bullet points.
- The definitions of the different terms presented in Table 1 were indeed paraphrased, as pointed out by the reviewer. The reason was that we considered several references and wanted to summarise these which made the definitions prone to (mis)interpretation. The revised version directly quotes relevant authors for each keyword, and we have analysed them in the discussion below the table. We believe that this has added value to this subsection.
- In this latest revised article, Table 1 includes an etymology analysis of each keyword. This allows to understand where each word comes from and add a quantitative quality to the definitions.
- The discussion of Table 1 has been updated to reflect the changes made within Table 1.
- The keywords that were considered have been changed: biotechnology has been removed, ecomimicry and ecomimesis have been analysed together as they are too close in meaning and not referenced enough in literature, and biomorphism has been analysed together with biomorphic as a result to the bibliometric analysis’ update.
We would also like to stress that the definitions indeed are similar to an extent, relate to biomimicry and share overlapping elements. To showcase the differences and similarities between the words, it is important to reflect on these elements. Moreover, the goal of this paper is to have a better and overall understanding of BIA concerning terms and interpretations, and their applications and approaches. Many researchers use distinct biomimicry-related keywords which can lead to confusion.
Fig.2. lucks consistency as well. Presented scheme is not supported by thorough literature analysis. I said it already in previous version - from this scheme one could conclude that all the examples of biomorphism is biomimicry but it is not - by far. Most biomorphism examples are not biomimicry. The same goes for other terms in this scheme - there is no added value to this scheme - just compilation of "related" words. Even if it refers to another literature source, it is misleading. I would avoid this scheme until authors can add significant depth to it. Biomimicry institue has made it much more clear Venn diagram, maybe you could elaborate based on that. https://biomimicry.org/what-is-biomimicry/.
We thank the reviewer for pointing this out. We do believe, however, that the interpretations of the definitions bring insight to what nature-inspired design is under the different terms used. After thorough considerations and further studies, we have decided to delete the Venn diagram to avoid confusion and have expanded Table 1 with more references, as well as the explanation below the table.
I value the effort, however bibliometric analysis is performed very poorly.
We thank the reviewer for pointing this out and have updated the bibliometric analysis needed within the scope of this paper to quantitatively portray the terminology related to biomimicry. The changes are explained below.
- Proper syntax of the search including "and", "or", "not" (or other) and sensitivity to the word endings has to be prepared and described.
We have included the syntax in Table 2, to better represent the strings used in Scopus.
- Biomimetic and Biomimetics - why is this worth looking separately to these two words? Most probably all the "biomimetic" papers are included in "Biomimetics" search
This is indeed a valid observation and we have changed the string in Scopus to ‘Biomimetic OR Biomimetics’.
- Biomorphism search has only 22 results because the search is not accurate - it does not include varieties in the word endings - biomorphic, for instance
By the same token as bullet point 2., we have changed the string to ‘Biomorphism OR Biomorphic’, which resulted in a more representative result of 1044 contributions.
- Organic design 100k results- how those are relevant to the topic?
As mentioned in the introduction of subsection 2.2.1., the article elaborates on the terms that use nature as a generator for design. Organic design is an example of this and important to explain, in order to showcase the difference between the terminologies, and in particular, between BIA. The reason behind the high number of publications has been added in the discussion, including an additional reference. (Organic design was used by famous architect Frank Lloyd Wright in the early 1900s, which led to more publications from an early date.)
- Ecomimicry gives <10 results and is excluded from bibliometric analysis, but is included in the fig 2
We have deleted Figure 2 (Venn diagram) and amended the bibliometric analysis. ‘Ecomimicry’ and ‘Ecomimesis’ are conjointly included in Table 2 (bibliometric analysis).
- table 4 highlights the lack of deep analysis - it just shows that there is a lot of mentioning of some key - words across different fields. Because the search is not tuned, it is too wide
This is a valid observation and after reflecting on the focus of the paper, we have decided to amend the bibliometric research. While we believe that the bibliometric research is crucial in this section to quantify the results, we have deleted the tables that illustrated the geographical spread and the study area. Accordingly, we have updated Table 2 (publications from 1997 to 2022) and updated the text, including the research conducted of the geographical spread and the study area.
Table 5 - the same as for table 1 - it seems like interpretations not substantially different definitions.
We agree with the reviewer that this is prone to misunderstandings and after extensive considerations, we have decided to delete Table 5. Accordingly, we have decided to briefly elaborate on the different interpretations of several researchers and scientific papers within the main text, to better understand how biomimicry is perceived and interpreted among scholars.
In section 3 table 6 - keywords are again authors' interpretations because there are other literature sources where other keywords are used for the same example.
We thank the reviewer for this comment. It is indeed true that many scholars use various studies and interpret each case study differently. We would also like to remined the reviewer that the linked keywords are based on several articles, references and definitions analysed in Section 2. However, undeniably, there is some degree of interpretation involved, in our research but also in others. This can be observed in the classification of the Lotus Temple. While Sommese et al. [11] classify it as biomimicry, the Biomimicry Institute [25] and Chayaamor-Heil and Hannachi-Belkadi [23] deny this by stating that biomorphism is not at all a part of biomimicry, and that the Lotus Temple is thus not relevant in this article. Therefore, we would like to remind the reviewer that this has been discussed in 5. Discussion and contributes to the scope of this paper by highlighting these differences and similarities among scholars in terms of interpretations.
If authors want to keep analysis of definitions it HAS to be grounded in some kind of quantifiable parameters or otherwise prove that there is substantial difference.
Section 4 is better
Good luck!
We thank the reviewer for the comments and believe that the changes and clarifications regarding the analysis of the definitions in Section 2 are now better and have enhanced the quality of this paper. We would also like to thank the referee for the time spend reviewing our paper. We hope that the changes made meet your expectations.

Reviewer 2 Report
NA
Author Response
We thank the reviewer for accepting the manuscript for publishing. Therefore, we only changed the manuscript according to the comments of Reviewer 1.